# EDITABLE GRAPH NEURAL NETWORK FOR NODE CLASSIFICATIONS

## ABSTRACT

Although Graph Neural Networks (GNNs) have achieved significant success in various graph-based learning tasks, such as credit risk assessment in financial networks and fake news detection in social networks, the trained GNNs can still make errors. These errors can potentially have a severe negative impact on society. *Model editing*, which corrects the model behaviour on wrongly predicted target samples while leaving model predictions unchanged on unlated samples, has garnered significant interest in the fields of computer vision and natural language processing. However, model editing for graph neural networks (GNNs) is rarely explored, despite GNNs' widespread applicability. To fill the gap, we first observe that existing model editing methods significantly deteriorate prediction accuracy (up to $50\%$ accuracy drop) in GNNs while a slight accuracy drop in multi-layer perception (MLP). The rationale behind this observation is that the node aggregation in GNNs will spread the editing effect throughout the whole graph. This propagation pushes the node representation far from its original one. Motivated by this observation, we propose Editable Graph Neural Networks (EGNN), a neighbor propagation-free approach to correct the model prediction on misclassified nodes. Specifically, EGNN simply stitches an MLP to the underlying GNNs, where the weights of GNNs are frozen during model editing. In this way, EGNN disables the propagation during editing while still utilizing the neighbor propagation scheme for node prediction to obtain satisfactory results. Experiments demonstrate that EGNN outperforms existing baselines in terms of effectiveness (correcting wrong predictions with lower accuracy drop), generalizability (correcting wrong predictions for other similar nodes), and efficiency (low training time and memory) on various graph datasets.

## 1 INTRODUCTION

Graph Neural Networks (GNNs) have achieved prominent results in learning feature and topology of graph data (Ying et al., 2018; Hamilton et al., 2017; Ling et al., 2023b; Zeng et al., 2020; Hu et al., 2020; Zhou et al.; Jiang et al., 2022a; Han et al., 2022a; Ling et al., 2023a). Based on spatial message passing, GNNs learn each node through aggregating representations of its neighbors and the node itself recursively. Once trained, the model is typically deployed as static artifacts to make decisions on a wide range of tasks, such as credit risk assessment in financial networks (Petrone & Latora, 2018), fake news detection in social networks (Shu et al., 2017), or prediction of drug-target interactions (Zhang et al., 2022). Although advanced GNN models can often deliver promising *overall performance* (e.g., classification accuracy across all test cases) on such datasets, in reality, **one particular wrong output can be a lot more damaging than another; therefore, some special interventions are required to ensure the outputs of certain high-stake instances to be correct or favorable.** For Computer Vision (CV) tasks, this can be to correctly identify a street-crossing child in front of a self-driving car. In Natural Language Processing (NLP) tasks, this can be to ensure an LLM-powered chatbot does not give out criminal advice. While graph data are often less intuitive than text and images, given the prevalence of graph learning applications under the abovementioned (and many more) high-stake scenarios, it would be irrefutably vital to ensure a drug discovery model will not produce toxic products; or to prevent a fraud prevention system from losing focus on its most vulnerable victim groups and cost them their life savings[1]

---

[1] According to FBI Internet Crime Report 2020, around 66% of the tech support fraud victims are over 60 years; yet, they are bearing at least 84% of the total losses (>$116 million). This suggests senior citizens are

Moreover, from a model maintenance perspective, **high-profile failure cases often manifest in a streaming manner** *after* **the initial model development (e.g., training)**, but during the actual user-facing deployment. Model editing serves as a way to **timely and efficiently deliver a guaranteed patch** for those post-hoc discovered errors, making it a practical tool for launching safer AI products.

Ideally, it is desirable to correct these serious errors (and generalize corrections to similar mistakes), while preserving the model's prediction accuracy on unrelated input samples. To obtain generalization ability for similar samples, the most prevalent method is to fine-tune the model with a new label on the single example to be corrected. However, this approach often spoils the model prediction on other unrelated samples. To cope with the challenge, many model editing frameworks have been proposed to adjust model behaviors by correcting errors as they appear (Sinitsin et al., 2020a; Mitchell et al., 2021; 2022; De Cao et al., 2021). Specifically, these editors usually require an additional training phase to help the model "prepare" for the editing process before applying any edits (Sinitsin et al., 2020a; Mitchell et al., 2021; 2022; De Cao et al., 2021).

**Although model editing has shown promise to modify vision and language models, to the best of our knowledge, there is no existing work tackling the critical mistakes in graph data.** Despite the straightforward concept, it is challenging to efficiently change GNNs' behaviors on the massively connected nodes. First, due to the message-passing mechanism in GNNs, editing the model behavior on a single node can propagate changes across the entire graph, significantly altering the node's original representation, which may destroy the prediction performance on the training dataset. Therefore, compared to the neural networks for computer vision or natural language processing, it is harder to maintain the model prediction on other input samples. Second, unlike other types of neural networks, the input nodes are connected in the graph domain. Thus, when editing the model prediction on a single node using gradient descent, the representation of each node in the whole graph is required (Liu et al., 2022b; Han et al., 2023b; Hamilton et al., 2017). This distinction introduces complexity and computational challenges when editing GNNs, especially on large graphs.

In this work, we delve into studying the graph model editing problem, which is more challenging than the independent sample edits. We first observe the existing editors significantly harm the overall node classification accuracy although the misclassified nodes are corrected. The test accuracy drop is up to 50%, which prevents GNNs from being practically deployed. We experimentally study the rationale behind this observation from the lens of loss landscapes. Specifically, we visualize the loss landscape of the Kullback-Leibler (KL) divergence between node embeddings obtained before and after the model editing process in GNNs. We found that a slight weight perturbation can significantly enlarge the KL divergence. In contrast, other types of neural networks, such as Multi-Layer Perceptrons (MLPs), exhibit a much flatter region of the KL loss landscape and display greater robustness against weight variations. Such observations align with our viewpoint that after editing on misclassified samples, GNNs are prone to widely propagating the editing effect and affecting the remaining nodes.

Based on the sharp loss landscape, we propose Editable Graph Neural Network (`EGNN`), a neighbor propagation-free approach to correct the model prediction on the graph data. Specifically, suppose we have a well-trained GNN and we want to correct its prediction on some of the misclassified nodes. `EGNN` stitches a randomly initialized MLP to the trained GNN. We then train the MLP for a few iterations to ensure that it does not significantly alter the model's prediction. When performing the edit, we only update the parameter of the stitched MLP while freezing the parameter of GNNs during the model editing process. In particular, the node embeddings from GNNs are first inferred offline. Then MLP learns an additional representation, which is then combined with the fixed embeddings inferred from GNNs to make the final prediction. When a misclassified node is received, the gradient is back propagated to update the parameters of MLP instead of GNNs'. In this way, we decouple the *neighbor propagation process* of learning the structure-aware node embeddings from the *model editing process* of correcting the misclassified nodes. Thus, `EGNN` disables the propagation during editing while still utilizing the neighbor propagation scheme for node prediction to obtain satisfactory results. Compared to directly applying the existing model editing methods to GNNs:

- We can leverage the GNNs' structure learning meanwhile avoiding the spreading edition errors to guarantee the overall node classification task.

---

more likely to experience a severe financial setback due to being the victim of the said crime, making them a prioritized focus for a proper fraud protection system. This real-world example perfectly illustrates the fact that while predicting two different potential fraud victims is considered equal under some metrics valuing *overall performance*, the difference in real-life impact can be drastic.

- The experimental results validate our solution which could address all the erroneous samples and deliver up to **90% improvement in overall accuracy**.
- Via freezing GNNs' part, EGNN is scalable to address misclassified nodes in the million-size graphs. We save more than $2\times$ in terms of memory footprint and model editing time.

## 2 PRELIMINARY

**Graph Neural Networks.** Let $\mathcal{G} = (\mathcal{V}, \mathcal{E})$ be an undirected graph with $\mathcal{V} = (v_1, \cdots, v_{|\mathcal{V}|})$ and $\mathcal{E} = (e_1, \cdots, e_{|\mathcal{E}|})$ being the set of nodes and edges, respectively. Let $\boldsymbol{X} \in \mathbb{R}^{|\mathcal{V}| \times d}$ be the node feature matrix. $\boldsymbol{A} \in \mathbb{R}^{|\mathcal{V}| \times |\mathcal{V}|}$ is the graph adjacency matrix, where $\boldsymbol{A}_{i,j} = 1$ if $(v_i, v_j) \in \mathcal{E}$ else $\boldsymbol{A}_{i,j} = 0$. $\tilde{\boldsymbol{A}} = \tilde{\boldsymbol{D}}^{-\frac{1}{2}}(\boldsymbol{A} + \boldsymbol{I})\tilde{\boldsymbol{D}}^{-\frac{1}{2}}$ is the normalized adjacency matrix, where $\tilde{\boldsymbol{D}}$ is the degree matrix of $\boldsymbol{A} + \boldsymbol{I}$. In this work, we are mostly interested in the task of node classification, where each node $v \in \mathcal{V}$ is associated with a label $y_v$, and the goal is to learn a representation $\boldsymbol{h}_v$ from which $y_v$ can be easily predicted. To obtain such a representation, GNNs follow a neural message passing scheme (Kipf & Welling, 2017). Specifically, GNNs recursively update the representation of a node by aggregating representations of its neighbors. For example, the $l^{\text{th}}$ Graph Convolutional Network (GCN) layer (Kipf & Welling, 2017) can be defined as:

$$\boldsymbol{H}^{(l+1)} = \text{ReLU}(\tilde{\boldsymbol{A}}\boldsymbol{H}^{(l)}\boldsymbol{\Theta}^{(l)}), \tag{1}$$

where $\boldsymbol{H}^{(l)}$ is the node embedding matrix containing the $\boldsymbol{h}_v$ for each node $v$ at the $l^{\text{th}}$ layer and $\boldsymbol{H}^{(0)} = \boldsymbol{X}$. $\boldsymbol{\Theta}^{(l)}$ is the weight matrix of the $l^{\text{th}}$ layer.

**The Model Editing Problem.** The goal of model editing is to alter a base model's output for misclassified sample $x_e$ as well as its similar samples via model finetuning only using a single pair of input $x_e$ and desired output $y_e$ while leaving model behavior on unrelated inputs intact (Sinitsin et al., 2020a; Mitchell et al., 2021; 2022). We are the first to propose the model editing problem in graph data, where the decision faults on a small amount of critical nodes can lead to significant financial loss and/or fairness concerns. For the node classification, suppose a well-trained GNN incorrectly predicts a specific node. **Model editing** is used to correct the undesirable prediction behavior for that node by using the node's features and desired label to update the model. Ideally, the model editing ensures that the updated model makes accurate predictions for the specific node and its similar samples while maintaining the model's original behavior for the remaining unrelated inputs. Some model editors, such as the one presented in this paper, require a training phase before they can be used for editing.

## 3 PROPOSED METHODS

In this section, we first empirically show vanilla model editing performs extremely worse for GNNs compared with MLPs due to node propagation (Section 3.1). Intuitively, due to the message-passing mechanism in GNNs, editing the model behavior on a single node can propagate changes across the entire graph, significantly altering the node's original representation. Then through visualizing the loss landscape, we found that for GNNs, even a slight weight perturbation, the node representation will be far away from the original one (Section 3.2). Based on the observation, we propose a propagation-free GNN editing method called EGNN (Section 3.3).

### 3.1 MOTIVATION: MODEL EDITING MAY CRY IN GNNS

**Setting:** We train GCN, GraphSAGE, and MLP on Cora, Flickr, Reddit, and ogbn-arxiv, respectively, following the training setup as described in Section 5. To evaluate the difficulty of editing, *we ensured that the node to be edited was not present during training*, meaning that *the models were trained inductively*. Specifically, we trained the model on a subgraph containing only the training node and evaluated its performance on the validation and test set of nodes. Next, we selected a misclassified node from the validation set and applied gradient descent only on that node until the model made a correct prediction for it. Following previous work (Sinitsin et al., 2020a; Mitchell et al., 2022), we perform 50 independent edits and report the averaged test accuracy before and after performing a single edit.

**Results:** As shown in Table 1, we observe that **(1)** GNNs consistently outperform MLP on all the graph datasets before editing. This is consistent with the previous graph analysis results, where the neural message passing involved in GNNs extracts the graph topology to benefit the node representation learning and thereby the classification accuracy. **(2)** After editing, the accuracy drop of GNNs is significantly larger than that of MLP. For example, GraphSAGE has an almost 50% drop in test accuracy on ogbn-arxiv after editing even a single point. MLP with editing even delivers higher overall accuracies on Flickr and ogbn-arxiv compared with GNN-based approaches. One

Table 1: The test accuracy (%) before ("w./o. edit") and after editing ("w./ edit") on one single data point. $\Delta$ Acc is the accuracy drop before and after performing the edit. All results are averaged over 50 simultaneous model edits. The best result is highlighted by **bold faces.**

|           |            | GCN     | GraphSAGE | MLP      |
|-----------|------------|---------|-----------|----------|
|           | w./o. edit | **89.4**  | 86.6      | 71.8     |
| Cora      | w./ edit   | **84.36** | 82.06     | 68.33    |
|           | $\Delta$ Acc. | 5.03↓   | 4.53↓     | **3.46** ↓ |
|           | w./o. edit | **51.19** | 49.03     | 46.77    |
| Flickr    | w./ edit   | 13.94   | 17.15     | **36.68**  |
|           | $\Delta$ Acc. | 37.25↓  | 31.88↓    | **10.08** ↓ |
|           | w./o. edit | 95.52   | **96.55**   | 72.41    |
| Reddit    | w./ edit   | **75.20** | 55.85     | 69.86    |
|           | $\Delta$ Acc. | 20.32↓  | 40.70↓    | **2.54** ↓ |
|           | w./o. edit | **70.20** | 68.38     | 52.65    |
| ogbn-arxiv | w./ edit  | 23.70   | 19.06     | **45.15**  |
|           | $\Delta$ Acc. | 46.49↓  | 49.31↓    | **7.52**↓  |

of the intuitive explanations is the slightly fine-tuned weights in MLP mainly affect the target node, instead of other unrelated samples. However, due to the message-passing mechanism in GNNs, the edited node representation can be propagated over the whole graph and thus change the decisions on a large area of nodes. These comparison results reveal the unique challenge in editing the correlated nodes with GNNs, compared with the conventional neural networks working on isolated samples. **(3)** After editing, the test accuracy of GCN, GraphSAGE, and MLP become too low to be practically deployed. This is quite different to the model editing problems in computer vision and natural language processing, where the modified models only suffer an acceptable accuracy drop.

## 3.2 SHARP LOCALITY OF GNNS THROUGH LOSS LANDSCAPE

Intuitively, due to the message-passing mechanism in GNNs, editing the model behavior for a single node can cause the editing effect to propagate across the entire graph. This propagation pushes the node representation far from its original one. **Thus, we hypothesized that the difficulty in editing GNNs as being due to the neighbor propagation of GNNs.** The model editing aims to correct the prediction of the misclassified node using the cross-entropy loss of desired label. Intuitively, the large accuracy drop can be interpreted as the low model prediction similarity before and after model editing, named as the locality.

To quantitatively measure the locality, we use the metric of KL divergence between the node representations learned before and after model editing. The higher KL divergence means after editing, the node representation is far away from the original one. In other words, the higher KL divergence implies poor model locality, which is undesirable in the context of model editing. Particularly, we visualize the locality loss landscape for Cora dataset in Figure 1. We observe several **insights:** (1) GNNs (e.g., GCN and GraphSAGE) suffer from a much sharper loss landscape. Even slightly editing the weights, KL divergence loss is dramatically enhanced. That means GNNs are hard to be fine-tuned while keeping the locality. (2) MLP shows a flatter loss landscape and demonstrates much better locality to preserve overall node representations. This is consistent to the accuracy analysis in Table 1, where the accuracy drop of MLP is smaller. To deeply understand why model editing fails to work in GNNs, we also provide a pilot theoretical analysis on the KL locality difference between before/after model editing for one-layer GCN and MLP in Appendix E. **We theoretically show that when model editing corrects the model predictions on misclassified nodes, GNNs are susceptible to altering the predictions on other connected nodes. This phenomenon results in an increased KL divergence difference.**

Specifically, in Appendix E, we analyze KL locality loss via Talor expansion in model weight space, where model weight variation is determined by cross-entropy loss of the target sample. We found that KL locality loss is related to node feature similarity score. Consequently, we transform the node feature similarity score as the distance between the node feature matrix $\mathbf{X}$ and the subspace $\mathcal{M}$ where

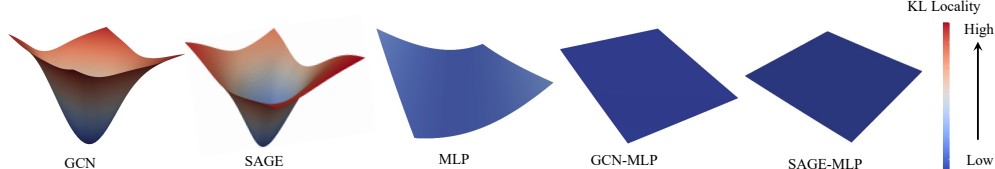

Figure 1: The loss landscape of various model architectures on Cora dataset. Similar results can be found in Appendix D

all row vectors are equivalent. Based on the spectral theory of graph, we can derive that the distance between aggregated node feature $\tilde{\mathbf{A}}\mathbf{X}$ and space $\mathcal{M}$ is higher than that of the original feature $\mathbf{X}$.

### 3.3 EGNN: Neighbor Propagation Free GNN Editing

In our previous analysis, we hypothesized that the difficulty in editing GNNs as being due to the neighbor propagation. However, as Table 1 suggested, the neighbor propagation is necessary for obtaining good performance on graph datasets. On the other hand, MLP could stabilize most of the node representations during model editing although it has worse node classification capability. Thus, we need to find a way to "disable" the propagation during editing while still utilizing the neighbor propagation scheme for node prediction to obtain satisfactory results. Following the motivation, we propose to combine a compact MLP to the well-trained GNN and only modify the MLP during editing. In this way, we can correct the model's predictions through this additional MLP while freezing the neighbor propagation. Meanwhile during inference, both the GNN and MLP are used together for prediction in tandem to harness the full potential of GNNs for prediction. The whole algorithm is shown in Algorithm 1.

---

**Algorithm 1:** Proposed EGNN

---

**procedure** MLP TRAINING PROCEDURE:

    **Input:** MLP $g_{\mathbf{\Phi}}$, dataset $\mathcal{D}$, the node embedding $\boldsymbol{h}_v$ for each node $v$ in $\mathcal{D}$

    **for** $t = 1, \cdots, T$ **do**

        Sample $\boldsymbol{x}_v, y_v \sim \mathcal{D}^{\text{train}}$

        $\mathcal{L}_{\text{loc}} = \text{KL}(\boldsymbol{h}_v + g_{\mathbf{\Phi}}(\boldsymbol{x}_v) \| \boldsymbol{h}_v)$

        $\mathcal{L}_{\text{task}} = -\log p_{\mathbf{\Phi}}(y_v | \boldsymbol{h}_v + g_{\mathbf{\Phi}}(\boldsymbol{x}_v))$

        $\mathcal{L} = \mathcal{L}_{\text{task}} + \alpha \mathcal{L}_{\text{loc}}$

        $\mathbf{\Phi} \leftarrow Adam(\mathbf{\Phi}, \nabla\mathcal{L})$

    **end**

**end**

**procedure** EGNN EDIT PROCEDURE:

    **Input:** data pair $x_e, y_e$ to be edited, the node embedding $\boldsymbol{h}_e$ for node $e$

    $\hat{y} = \arg\max_y p_{\mathbf{\Phi}}(y | \boldsymbol{x}_e, \boldsymbol{h}_e)$

    **while** $\hat{y} \neq y_e$ **do**

        $\mathcal{L} = -\log p_{\mathbf{\Phi}}(y | \boldsymbol{x}_e, \boldsymbol{h}_e)$

        $\mathbf{\Phi} \leftarrow Adam(\mathbf{\Phi}, \nabla\mathcal{L})$

    **end**

**end**

---

**Before editing.** We first stitch a randomly initialized compact MLP to the trained GNN. We freeze the weights of GNN in this step. To mitigate the potential impact of random initialization on the model's prediction, we introduce a training procedure for the stitched MLP, as outlined in Algorithm 1 "MLP TRAINING PROCEDURE": we train the MLP for a few iterations to ensure that it does not significantly alter the model's prediction. By freezing GNN's weights, we first get the node embedding $\boldsymbol{h}_v$ at the last layer of the trained GNN by running a single forward pass. We then stitch the MLP with the trained GNNs. Mathematically, we denote the MLP as $g_{\mathbf{\Phi}}$ where $\mathbf{\Phi}$ is the parameters of MLP. For a given input sample $\boldsymbol{x}_v, \boldsymbol{y}_v$, the model output now becomes $\boldsymbol{h}_v + g_{\mathbf{\Phi}}(\boldsymbol{x}_v)$. We calculate two loss based on the prediction, i.e., the task-specific loss $\mathcal{L}_{\text{task}}$ and the locality loss $\mathcal{L}_{\text{loc}}$. Namely,

$$\mathcal{L}_{\text{task}} = -\log p_{\mathbf{\Phi}}(y_v | \boldsymbol{h}_v + g_{\mathbf{\Phi}}(\boldsymbol{x}_v)),$$
$$\mathcal{L}_{\text{loc}} = \text{KL}(Softmax(\boldsymbol{h}_v + g_{\mathbf{\Phi}}(\boldsymbol{x}_v)) \| Softmax(\boldsymbol{h}_v)),$$

where $\boldsymbol{h}_v + g_{\boldsymbol{\Phi}}(\boldsymbol{x}_v)$ is the model prediction with the additional MLP and $p_{\boldsymbol{\Phi}}(y_v|\boldsymbol{h}_v + g_{\boldsymbol{\Phi}}(\boldsymbol{x}_v))$ is the probability of class $y_v$ given by the model. $\mathcal{L}_{\text{task}}$ is the cross-entropy between the model prediction and label. $\mathcal{L}_{\text{loc}}$ is the locality loss, which equals KL divergence between the original prediction $\boldsymbol{h}_v$ and the prediction with the additional MLP $\boldsymbol{h}_v + g_{\boldsymbol{\Phi}}(\boldsymbol{x}_v)$. The final loss $\mathcal{L}$ is the weighted combination of two parts, i.e., $\mathcal{L} = \mathcal{L}_{\text{task}} + \alpha \mathcal{L}_{\text{loc}}$ where $\alpha$ is the weight for the locality loss. $\mathcal{L}$ is used to guide the MLP to fit the task while keep the model prediction unchanged. **We present the ablation study on the $\mathcal{L}_{\textbf{task}}$ and $\mathcal{L}_{\textbf{loc}}$ in Appendix D.5.**

**When editing. `EGNN` freezes the model parameters of GNN and only updates the parameters of MLP.** Specifically, as outlined in Algorithm 1 "EGNN EDIT PROCEDURE", we update the parameters of MLP until the model prediction for the misclassified sample is corrected. Since MLP only relies on the node features, we can easily perform these updates in mini-batches, which enables us to edit GNNs on large graphs. Lastly, we visualize the KL locality loss landscape of EGNN (including GCN-MLP and SAGE-MLP) in Figure 1. It is seen that the proposed EGNN shows the most flattened loss landscape than MLP and GNNs, which implied that EGNN can preserve overall node representations better than other model architectures.

## 4 RELATED WORK AND DISCUSSION

Due to the page limit, below we discuss the related work on model editing. We also discuss the limitation in Appendix B.

**Model Editing.** Many approaches have been proposed for model editing. The most straightforward method adopts standard fine-tuning to update model parameters based on misclassified samples while preserving model locality via constraining parameters travel distance in model weight space (Zhu et al., 2020; Sotoudeh & Thakur, 2019). Work (Sinitsin et al., 2020b) introduces meta-learning to find a pre-trained model with rapid and easy finetuned ability for model editing. Another way to facilitate model editing relies on external learned editors to modify model editing considering several constraints (Mitchell et al., 2021; Hase et al., 2021; De Cao et al., 2021; Mitchell et al., 2022). The editing of the activation map is proposed to correct misclassified samples in (Dai et al., 2021; Meng et al., 2022) due to the belief of knowledge attributed to model neurons. While existing approaches either modify the base model parameters or introduce separate external modules to achieve desired prediction changes, they assume data to be independent and identically distributed (i.i.d.). This assumption might not hold well for graph data, given the fundamental node interactions that occur during neighborhood propagation. In this paper, we propose EGNN, using a stitched MLP module to edit the base GNN model, for node classification tasks. The key insight behind this solution is the sharp locality of GNNs, i.e., the prediction of GNNs can be easily altered after model editing.

## 5 EXPERIMENTS

The experiments are designed to answer the following research questions. **RQ1:** Can `EGNN` correct the wrong model prediction? Moreover, what is the difference in accuracy before and after editing using `EGNN` ? **RQ2:** Can the edits generalize to correct the model prediction on other similar inputs? **RQ3:** What is the time and memory requirement of `EGNN` to perform the edits?

### 5.1 EXPERIMENTAL SETUP

**Datasets and Models.** To evaluate `EGNN` , we adopt four small-scale and four large-scale graph benchmarks from different domains. For small-scale datasets, we adopt Cora, A-computers (Shchur et al., 2018), A-photo (Shchur et al., 2018), and Coauthor-CS (Shchur et al., 2018). For large-scale datasets, we adopt Reddit (Hamilton et al., 2017), Flickr (Zeng et al., 2020), *ogbn-arxiv* (Hu et al., 2020), and *ogbn-products* (Hu et al., 2020). We have incorporated `EGNN` with traditional GNN models such as GCN (Kipf & Welling, 2017) and GraphSAGE (Hamilton et al., 2017), as well as with GNNs that decouple propagation from the learning process, e.g., SGC (Wu et al., 2019) and SIGN (Frasca et al., 2020). *To avoid creating confusion, GCN and GraphSAGE are all trained with the whole graph at each step*. We evaluate `EGNN` under the **inductive setting**. Namely, we trained the model on a subgraph containing only the training node and evaluated it on the whole graph. Details about the hyperparameters and datasets are in Appendix A.

**Compared Methods.** We compare our EGNN editor with the following two baselines: the vanilla gradient descent editor (GD) and Editable Neural Network editor (ENN) (Sinitsin et al., 2020a). GD is the same editor we used in our preliminary analysis in Section 3. **We note that for other model editing, e.g., MEND (Mitchell et al., 2021), SERAC (Mitchell et al., 2022) are tailored for NLP applications, which cannot be directly applied to the graph area**. Specifically, GD applies the gradient descent **on the parameters of GNN** until the GNN makes right prediction. ENN trains **the parameters of GNN** for a few steps to make it prepare for the following edits. Then similar to GD editor, it applies the gradient descent **on the parameters of GNN** until the GNN makes right prediction. For EGNN , we only train **the stitched MLP** for a few steps. Then we only update **weights of MLP** during edits. Detailed hyperparameters are listed in Appendix A.

**Evaluation Metrics.** Following previous work (Sinitsin et al., 2020a; Mitchell et al., 2022; 2021), we evaluate the effectiveness of different methods by the following three metrics. **DrawDown (DD)**, which is the mean difference of test accuracy before and after performing an edit. A smaller drawdown indicates a better editor locality. **Success Rate (SR)**, which is defined as the rate of edits, where the editor successfully corrects the model prediction. **Edit Time**, which is defined as the wall-clock time of a single edit that corrects the model prediction.

## 5.2 THE EFFECTIVENESS OF EGNN IN EDITING GNNS

Table 2: The results on four small scale datasets after applying one single edit. The reported number is averaged over 50 independent edits. **SR** is the edit success rate, **Acc** is the test accuracy after editing, and **DD** are the test drawdown, respectively. "OOM" is the out-of-memory error.

| | Editor | Cora | | | A-computers | | | A-photo | | | Coauthor-CS | | |
|---|---|---|---|---|---|---|---|---|---|---|---|---|---|
| | | Acc↑ | DD↓ | SR↑ | Acc↑ | DD↓ | SR↑ | Acc↑ | DD↓ | SR↑ | Acc↑ | DD↓ | SR↑ |
| GCN | GD | 84.37±5.84 | 5.03±6.40 | 1.0 | 44.78±22.41 | 43.09±22.32 | 1.0 | 28.70±21.26 | 65.08±20.13 | 1.0 | 91.07±3.23 | 3.30±2.22 | 1.0 |
| | ENN | 37.16±3.80 | 52.24±4.76 | 1.0 | 15.51±10.99 | 72.36±10.87 | 1.0 | 16.71±14.81 | 77.07±15.20 | 1.0 | 4.94±3.78 | 89.43±3.34 | 1.0 |
| | EGNN | **87.80**±2.34 | **1.80**±2.13 | **1.0** | **82.85**±5.20 | **2.32**±5.11 | 0.98 | **91.97**±5.85 | **2.39**±5.34 | **1.0** | **94.54**±0.07 | **-0.17**±0.07 | **1.0** |
| Graph-SAGE | GD | 82.06±4.33 | 4.54±5.32 | 1.0 | 21.68±20.98 | 61.15±20.33 | 1.0 | 38.98±30.24 | 55.32±29.35 | 1.0 | 90.15±5.58 | 5.01±5.32 | 1.0 |
| | ENN | 33.16±1.45 | 53.44±2.23 | 1.0 | 16.89±16.98 | 65.94±16.75 | 1.0 | 15.06±11.92 | 79.24±11.25 | 1.0 | 13.71±2.73 | 81.45±2.11 | 1.0 |
| | EGNN | **85.65**±2.23 | **0.55**±1.26 | **1.0** | **84.34**±4.84 | **2.72**±5.03 | 0.94 | **92.53**±2.90 | **1.83**±3.22 | **1.0** | **95.27**±0.08 | **-0.01**±0.10 | **1.0** |

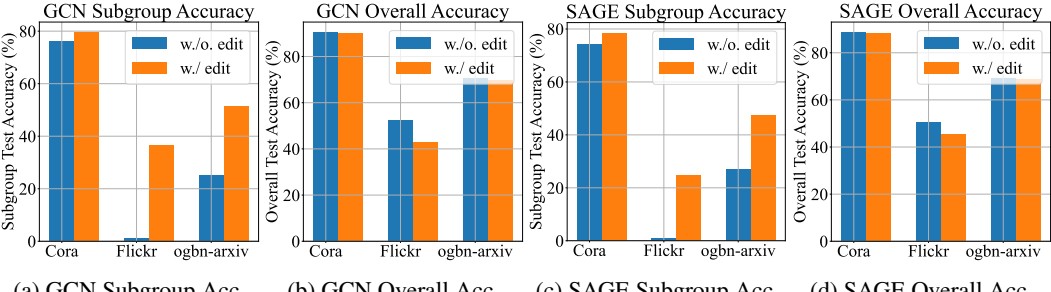

Figure 2: Sequential edit test drawdown of GCN and GraphSAGE on Reddit and ogbn-arxiv dataset. Due to the page limit, more similar results can be found in Appendix D.2.

(a) GCN Subgroup Acc. (b) GCN Overall Acc. (c) SAGE Subgroup Acc. (d) SAGE Overall Acc.

Figure 3: The subgroup and overall test accuracy before and after one single edit. The results are averaged over 50 independent edits.

In many real-world applications, it is common to encounter situations where our trained model produces incorrect predictions on unseen data. It is crucial to address these errors as soon as they are identified. To assess the usage of editors in real-world applications (**RQ1**), **we select misclassified nodes from the validation set, which is not seen during the training process.** Then we employ the editor to correct the model's predictions for those misclassified nodes, and measure the drawdown and edit success rate on the test set. The results after editing on a single node are shown in Table 2, Table 17, and Table 3. We observe:

❶ *Unlike editing Transformers on text data (Mitchell et al., 2021; 2022; Huang et al., 2023), all editors can successfully correct the model prediction in graph domain.* As shown in Table 3, all

editors have 100% success rate when edit GNNs. In contrast, for transformers, the edit success rate is often less than 50% and drawdown is much smaller than GNNs (Mitchell et al., 2021; 2022; Huang et al., 2023). This observation suggests that **unlike transformers, GNNs can be easily perturbed to produce correct predictions. However, at the cost of huge drawdown on other unrelated nodes. Thus, the main challenge lies in maintaining the locality between predictions for unrelated nodes before and after editing.** This observation aligns with our initial analysis, which highlighted the edit on a single node may propagate throughout the entire graph.

❷ *Even when the propagation is separated from the learning process, GNNs still face editing challenges arising from node mixing.* Some GNNs, such as SGC (Wu et al., 2019) and SIGN (Frasca et al., 2020), explicitly decouple neighbor propagation from the learning process, treating it as a preprocessing step. Intuitively, decoupling-based GNNs only take processed node features as inputs, eliminating the need for propagation. This raises an intriguing question: Do these GNNs still face the same editing challenges? Due to the page limit, we present the results in Appendix Table 17. Table 17 shows that decoupling-based GNNs still suffer. These results can be explained by the fact that the node features have been mixed due to the propagation-based preprocessing step. In this case, it is still hard to maintain the original representation after editing. To verify our hypothesis, we fed two different sets of features to the stitched MLP in EGNN . The first set comprised raw features without preprocessing, termed "EGNN (raw feat)" in Table 17 while the second incorporated processed features, labeled "EGNN (prop. feat.)". Our observations reveal that "EGNN (raw feat)" considerably outperforms its counterpart which utilizes propagated features in terms of drawdown. This ablation study firmly supports our claim that node propagation is a key factor for the graph editing challenge. Additionally, the experiments for various model architectures with different neighborhood aggregation strengths can be found in Appendix D.3. We also discuss the difference between our method and the adapter (Houlsby et al., 2019) in Appendix D.4.

Table 3: The results on four large scale datasets after applying one single edit. "OOM" is the out-of-memory error.

| | Editor | Flickr | | | Reddit | | | ogbn-arxiv | | | ogbn-products | | |
|---|---|---|---|---|---|---|---|---|---|---|---|---|---|
| | | Acc↑ | DD↓ | SR↑ | Acc↑ | DD↓ | SR↑ | Acc↑ | DD↓ | SR↑ | Acc↑ | DD↓ | SR↑ |
| GCN | GD | 13.95±11.0 | 37.25±10.2 | 1.0 | 75.20±12.3 | 20.32±11.3 | 1.0 | 23.71±16.9 | 46.50±14.9 | 1.0 | OOM | OOM | 0 |
| | ENN | 25.82±14.9 | 25.38±16.9 | 1.0 | 11.16±5.1 | 84.36±3.1 | 1.0 | 16.59±7.7 | 53.62±6.7 | 1.0 | OOM | OOM | 0 |
| | EGNN | **44.91±12.2** | **6.34±10.3** | **1.0** | **94.46±0.4** | **1.03±0.6** | **1.0** | **67.34±8.7** | **2.67±4.4** | **1.0** | **74.19±3.4** | **0.81±0.23** | **1.0** |
| Graph-SAGE | GD | 17.16±12.2 | 31.88±12.2 | 1.0 | 55.85±22.5 | 40.71±20.3 | 1.0 | 19.07±14.1 | 36.68±10.1 | 1.0 | OOM | OOM | 0 |
| | ENN | 28.73±5.6 | 20.31±5.6 | 1.0 | 5.88±3.9 | 90.68±4.3 | 1.0 | 8.14±8.6 | 47.61±7.6 | 1.0 | OOM | OOM | 0 |
| | EGNN | **43.52±10.8** | **5.12±10.8** | **1.0** | **96.50±0.1** | **0.05±0.1** | **1.0** | **67.91±2.9** | **0.64±2.3** | **1.0** | **76.27±0.6** | **0.17±0.10** | **1.0** |

❸ *EGNN significantly outperforms both GD and ENN in terms of the test drawdown.* This is mainly because both GD and ENN try to correct the model's predictions by updating the parameters of Graph Neural Networks (GNNs). This process inevitably relies on neighbor propagation. In contrast, EGNN has much better test accuracy after editing. Notably, for Reddit, the accuracy drop decreases from roughly 80% to ≈ 1%, which is significantly better than the baseline. This is because EGNN decouples the neighbor propagation with the editing process. Interestingly, ENN is significantly worse than the vanilla editor, i.e., GD, when applied to GNNs. As shown in Appendix D, we found that this discrepancy arises from the ENN training procedure, which significantly compromises the model's performance to prepare it for editing.

In Figure 6, 7, and 2 we present the ablation study under the sequential setting. This is a more challenging scenario where the model is edited sequentially as errors arise. In particular, we plot the test accuracy drawdown against the number of sequential edits for GraphSAGE on the ogbn-arxiv dataset. We observe that ❹ *EGNN consistently surpasses both GD and ENN in the sequential setting.* However, the drawdown is considerably greater than that in the single edit setting. For instance, EGNN exhibits a 0.64% drawdown for GraphSAGE on the ogbn-arxiv dataset in the single edit setting, which escalates up to a 20% drawdown in the sequential edit setting. These results also highlight the hardness of maintaining the locality of GNN prediction after editing.

### 5.3 THE GENERALIZATION OF THE EDITS OF EGNN

Ideally, we aim for the edit applied to a specific node to generalize to similar nodes while preserving the model's initial behavior for unrelated nodes. To evaluate the generalization of the EGNN edits, we conduct the following experiment:

**(1)** We first select a particular group (i.e., class) of nodes based on their labels. **(2)** Next, we randomly flip the labels of 10% of the training nodes within this group and train a GNN on the modified training

Table 4: The edit time and memory required for editing.

| | Editor | Flickr | | Reddit | | ogbn-arxiv | | ogbn-products | |
| --- | --- | --- | --- | --- | --- | --- | --- | --- | --- |
| | | Edit Time (ms) | Peak Memory (MB) | Edit Time (ms) | Peak Memory (MB) | Edit Time (ms) | Peak Memory (MB) | Edit Time (ms) | Peak Memory (MB) |
| GCN | GD | 379.86 | 707 | 1835.24 | 3429 | 663.17 | 967 | OOM | OOM |
| | EGNN | 246.63 | 315 | 765.15 | 2089 | 299.71 | 248 | 5122.53 | 5747 |
| Graph-SAGE | GD | 712.07 | 986 | 4781.92 | 5057 | 668.77 | 1109 | OOM | OOM |
| | EGNN | 389.37 | 328 | 1516.68 | 2252 | 174.82 | 260 | 5889.59 | 6223 |

set. **(3)** For each flipped training node, we correct the trained model's prediction for that node back to its original class and assess whether the model's predictions for other nodes in the same group are also corrected. If the model's predictions for other nodes in the same class are also corrected after modifying a single flipped node, it indicates that the `EGNN` edits can effectively generalize to address similar erroneous behavior in the model.

To answer **RQ2**, we conduct the above experiments and report the **subgroup and overall test accuracy** after performing a single edit on the flipped training node. The results are shown in Figure 3. We observe that: ❺ *From Figure 3a and Figure 3c, EGNN significantly improves the subgroup accuracy after performing even a single edit.* Notably, the subgroup accuracy is significantly lower than the overall accuracy. For example, on Flickr dataset, both GCN and GraphSAGE have a subgroup accuracy of less than 5% before editing. This is mainly because the GNN is trained on

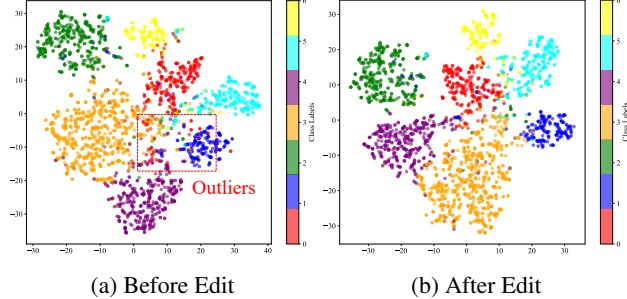

(a) Before Edit       (b) After Edit

Figure 4: T-SNE visualizations of GNN embeddings before and after edits on the Cora dataset. The flipped nodes are all from class 0, which is marked in red color.

the graph where 10% labels of the training node in the subgroup are flipped. However, even after editing on a single node, the subgroup accuracy is significantly boosted. These results indicate that the `EGNN` edits can effectively generalize to address the wrong prediction on other nodes in the same group. In Figure 4, we also visualize the node embeddings before and after editing by `EGNN` on the Cora dataset. We note that all of the flipped nodes are from class 0, which is marked in red color in Figure 4. Before editing, the red cluster has many outliers that lie in the embedding space of other classes. This is mainly because the labels of some of the nodes in this class are flipped. In contrast, after editing, the nodes in the red cluster become significantly closer to each other, with a substantial reduction in the number of outliers.

## 5.4 The Efficiency of EGNN

We want to patch the model as soon as possible to correct errors as they appear. Thus ideally, the editor should be efficient and scalable to large graphs. Here we summarize the edit time and memory required for performing the edits in Table 4. We observe that `EGNN` is about $2 \sim 5 \times$ faster than the GD editor in terms of the wall-clock edit time. This is because `EGNN` only updates the parameters of MLP, and totally gets rid of the expensive graph-based sparse operations (Liu et al., 2022b;a; Han et al., 2023b). Also, updating the parameters of GNNs requires storing the node embeddings in memory, which is directly proportional to the number of nodes in the graph and can be exceedingly expensive for large graphs However, with `EGNN` , we only use node features for updating MLPs, meaning that memory consumption is not dependent on the graph size. Consequently, `EGNN` can efficiently scale up to handle graphs with millions of nodes, e.g., ogbn-products, whereas the vanilla editor raises an OOM error.

## 6 Conclusion

We explore a and important problem, i.g., GNNs model editing for node classification. We empirically and theoretically show that the vanilla model editing method may not perform well due to node aggregation. Furthermore, we propose EGNN to correct misclassified samples while preserving other intact nodes, via stitching a trainable MLP. In this way, the power of GNNs for prediction and the editing-friendly MLP can be integrated together in EGNN.

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

# A EXPERIMENTAL SETTING

## A.1 DATASETS FOR NODE CLASSIFICATION

The details of datasets used for node classification are listed as follows:

- Cora (Sen et al., 2008) is the citation network. The dataset contains 2,708 publications with 5,429 links, and each publication is described by a 1,433-dimensional binary vector, indicating the presence or absence of corresponding words from a fixed vocabulary.

- A-computers (Shchur et al., 2018) is the segment of the Amazon co-purchase graph, where nodes represent goods, edges indicate that two goods are frequently bought together, node features are bag-of-words encoded product reviews.

- A-photo (Shchur et al., 2018) is similar to A-computers, which is also the segment of the Amazon co-purchase graph, where nodes represent goods, edges indicate that two goods are frequently bought together, node features are bag-of-words encoded product reviews.

- Coauthor-CS (Shchur et al., 2018) is the co-authorship graph based on the Microsoft Academic Graph from the KDD Cup 2016 challenge 3. Here, nodes are authors, that are connected by an edge if they co-authored a paper; node features represent paper keywords for each author's papers, and class labels indicate most active fields of study for each author.

- Reddit (Hamilton et al., 2017) is constructed by Reddit posts. The node in this dataset is a post belonging to different communities.

- *ogbn-arxiv* (Hu et al., 2020) is the citation network between all arXiv papers. Each node denotes a paper and each edge denotes citation between two papers. The node features are the average 128-dimensional word vector of its title and abstract.

- *ogbn-prducts* (Hu et al., 2020) is Amazon product co-purchasing network. Nodes represent products in Amazon, and edges between two products indicate that the products are purchased together. Node features are low-dimensional representations of the product description text.

Table 5: Statistics for datasets used for node classification.

| Dataset | # Nodes. | # Edges | # Classes | # Feat | Density |
|---|---|---|---|---|---|
| Cora | 2,485 | 5,069 | 7 | 1433 | 0.72‰ |
| A-computers | 13,381 | 245,778 | 10 | 767 | 2.6‰ |
| A-photo | 7,487 | 119, | 8 | 745 | 4.07‰ |
| Coauthor-CS | 18,333 | 81,894 | 15 | 6805 | 0.49‰ |
| Flickr | 89,250 | 899,756 | 7 | 500 | 0.11‰ |
| Reddit | 232,965 | 23,213,838 | 41 | 602 | 0.43‰ |
| *ogbn-arxiv* | 169,343 | 1,166,243 | 40 | 128 | 0.04‰ |
| *ogbn-products* | 2,449,029 | 61,859,140 | 47 | 218 | 0.01‰ |

## A.2 BASELINES FOR NODE CLASSIFICATION

We present the details of the hyperparameters of GCN, GraphSAGE, and the stitched MLP modules in Table 6. We use the Adam optimizer for all these models.

## A.3 HARDWARE AND SOFTWARE CONFIGURATION

All experiments are executed on a server with 500GB main memory, two AMD EPYC 7513 CPUs. All experiments are done with a single NVIDIA RTX A5000 (24GB). The software and package version is specified in Table 7:

Table 6: Training configuration for employed models

| Model | Dataset | #Layers | #Hidden | Learning rate | Dropout | Epoch |
|---|---|---|---|---|---|---|
| GraphSAGE | Cora | 2 | 32 | 0.01 | 0.1 | 200 |
| | A-computers | 2 | 32 | 0.01 | 0.1 | 400 |
| | A-photo | 2 | 32 | 0.01 | 0.1 | 400 |
| | Coauthor-CS | 2 | 32 | 0.01 | 0.1 | 400 |
| | Flickr | 2 | 256 | 0.01 | 0.3 | 400 |
| | Reddit | 2 | 256 | 0.01 | 0.5 | 400 |
| | ogbn-arxiv | 3 | 128 | 0.01 | 0.5 | 500 |
| | ogbn-products | 3 | 256 | 0.002 | 0.5 | 500 |
| GCN | Cora | 2 | 32 | 0.01 | 0.1 | 200 |
| | A-computers | 2 | 32 | 0.01 | 0.1 | 400 |
| | A-photo | 2 | 32 | 0.01 | 0.1 | 400 |
| | Coauthor-CS | 4 | 32 | 0.01 | 0.1 | 400 |
| | Flickr | 2 | 256 | 0.01 | 0.3 | 400 |
| | Reddit | 2 | 256 | 0.01 | 0.5 | 400 |
| | ogbn-arxiv | 3 | 128 | 0.01 | 0.5 | 500 |
| | ogbn-products | 3 | 256 | 0.002 | 0.5 | 500 |
| MLP | Cora | 2 | 32 | 0.01 | 0.1 | 200 |
| | A-computers | 2 | 32 | 0.01 | 0.1 | 400 |
| | A-photo | 2 | 32 | 0.01 | 0.1 | 400 |
| | Coauthor-CS | 4 | 32 | 0.01 | 0.1 | 400 |
| | Flickr | 2 | 256 | 0.01 | 0.3 | 400 |
| | Reddit | 2 | 256 | 0.01 | 0.5 | 400 |
| | ogbn-arxiv | 3 | 128 | 0.01 | 0.5 | 500 |
| | ogbn-products | 3 | 256 | 0.002 | 0.5 | 500 |

Table 7: Package configurations of our experiments.

| Package | Version |
|---|---|
| CUDA | 11.3 |
| pytorch | 1.10.2 |
| torch-geometric | 1.7.2 |
| torch-scatter | 2.0.8 |
| torch-sparse | 0.6.12 |

## B  LIMITATIONS

Despite that `EGNN` is effective, generalized, and efficient, its main limitation is that currently, it will incur a larger inference latency, due to the extra MLP module. However, we note that this inference overhead is negligible. This is mainly because the computation of MLP only involve dense matrix operation, which is way more faster than the graph-based sparse operations Liu et al. (2022b;a); Han et al. (2023b).

## C  FUTURE WORK

There are abundant directions on top of our work, including (1) Enhancing the efficiency of editable graph neural networks training through various perspectives (Wang et al., 2018; Cazenavette et al., 2022; Jin et al., 2021; Feng et al., 2023; Han et al., 2022b) (e.g., model initialization, data, and gradient); (2) understanding why vanilla editable graph neural networks training fails from other perspectives (e.g., interpretation and information bottleneck) (Lundberg & Lee, 2017; Tishby et al., 2000; Liu et al., 2020); (3) Advancing the scalability, speed, and memory efficiency of editable graph neural networks training (Liu et al., 2022b;a; Han et al., 2023b); (4) Expanding the scope of editable training for other tasks (e.g., link prediction, and knowledge graph) (Lü & Zhou, 2011; Wang et al., 2014); (5) Investigating the potential issue concerning privacy, robustness, and fairness in the context

of editable graph neural networks training (Zheleva & Getoor, 2008; Jiang et al., 2023; Jin et al., 2020; Dai & Wang, 2021; Jiang et al., 2022b; Han et al., 2023a; Tang et al., 2020).

## D   MORE EXPERIMENTAL RESULTS

### D.1   MORE LOSS LANDSCAPE RESULTS

We visualize the locality loss landscape for Flickr dataset in Figure 5. Similarly, $Z$ axis denotes the KL divergence, X-Y axis is centered on the original model weights before editing and quantifies the weight perturbation scale after model editing. We observe similar observations: (1) GNNs architectures (e.g., GCN and GraphSAGE) suffer from a much sharper loss landscape at the convergence of original model weights. KL divergence locality loss is dramatically enhanced even for slight weights editing. (2) MLP shows a flatter loss landscape and demonstrates mild locality to preserve overall node representations, which is consistent with the accuracy analysis in Table 1. (3) The proposed EGNN shows the most flattened loss landscape than MLP and GNNs, which implied that EGNN can preserve overall node representations better than other model architectures.

We also provide an intuitive justification on loss landscape results. Firstly, both MLP and GCN-MLP are flat because they're exempt from neighborhood propagation. As for flatness comparison between MLP and GCN-MLP, we would like to note that the final output of GCN-MLP consists two parts: $\mathbf{h}_v$ from GCN, and $g_\Phi(\mathbf{x}_v)$ from MLP, where GCN part is fixed during editing the MLP parameters. In contrast, when perturbing the MLP model, the entire model is affected. Thus intuitively, when only editing the MLP parameters, the GCN-MLP model is flatter because it's harder to alter its final results when only a subpart (the MLP part) is perturbed, while in the case of the MLP, the whole model is susceptible to changes.

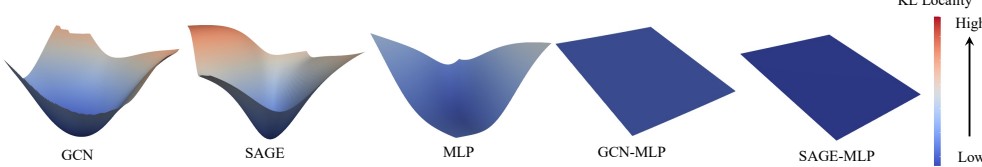

Figure 5: The loss landscape of various GNNs architectures on Flickr dataset.

### D.2   MORE SEQUENTIEL EDITING REUSLTS

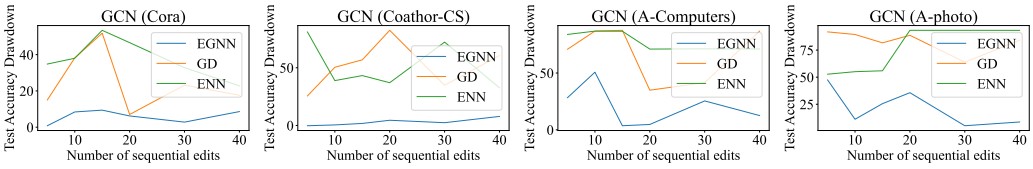

Figure 6: Sequential edit drawdown of GCN on four small scale datasets.

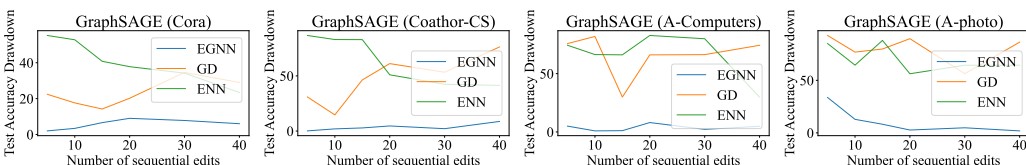

Figure 7: Sequential edit drawdown of GraphSAGE on four small scale datasets.

### D.3   EXPERIMENTS FOR MODEL ARCHITECTURES WITH DIFFERENT NEIGHBORHOOD PROPAGATION STRENGTHS

To investigate whether neighborhood propagation is the main difficulty in editing GNNs, we conduct experiments on various model architectures with different neighborhood propagation strengths.

Table 8: Test DrawDown (%) of different model architectures on five datasets.

| Dataset | GCN | GraphSAGE | SGC | SIGN | MLP |
|---|---|---|---|---|---|
| Cora | 5.03 | 4.53 | 3.73 | 4.80 | **3.46** |
| Flickr | 37.25 | 31.88 | 29.91 | 14.98 | **10.08** |
| A-computers | 43.09 | 61.15 | 60.97 | 27.27 | **6.98** |
| A-photo | 65.08 | 55.32 | 68.53 | 27.53 | **9.08** |
| Coauthor-CS | 3.30 | 5.01 | 3.13 | 1.42 | **0.57** |

Specifically, we choose the architectures conducting propagation in each layer (e.g., GCN, Graph-SAGE), conducting propagation only once (e.g., SGC and SIGN), and without propagation (i.e., MLP). The test drawdown (DD) for many architectures on various graph datasets is shown in Table 8. It is seen that SGC and SIGN are better than popular GNN models such as GCN and GraphSAGE. Yet, they significantly underperform compared to pure MLPs. This is mainly because the neighbor propagation in the preprocessing step still correlates to different input nodes. In a nutshell, model architectures with extensive neighborhood aggregation suffer more in model editing and demonstrate high test drawdown.

## D.4 ABLATION STUDY ON COMPARING EGNN WITH ADAPTER

Here we discuss the difference between `EGNN` and Adapter Houlsby et al. (2019). In Table 9 we compare `EGNN` against Adapter in terms of test drawdown using GCN. Below we list three key difference between them:

- The problems are different. Adapter tuning aims to adapt a pre-trained model to a new task with fewer parameters updated. Suppose we view correcting a GNN's wrong predictions as the new task, then adapter tuning does not try to maintain the GNN's predictions on unrelated inputs — even though this is one of the key goals for model editing.

- The architectures are different, and such difference matters in GNN editing. While both use MLP as the medium, adapter tuning inserts MLP modules between layers, where EGNN employs a single MLP parallel to the GNN. So in a GNN context, node features fed to such between-layer adapters are already mixed due to neighbor propagations, which is not ideal for edit robustness. The 'GCN + adapter' results below indicate a direct application of adapter tuning to GCN edit problem has a drawdown up to 80%, which is meaningless to the editing task.

- -The weight-update strategies are different, and again it matters. Since adapter tuning aims to adapt a model to a new task, it only employs one holistic fine-tuning process. In EGNN, we consider both task-specific and locality loss, designing a two-step approach. This design is noteworthy under model edit, as if we apply adapter tuning with $L_{task}$ and $L_{loc}$ as guidance, we can witness an improvement up to 60%. Though, such modified adapter tuning approach still significantly fell short of EGNN due to its reliance on neighbor propagation.

Table 9: Test DrawDown (the lower the better) comparison between EGNN and Adapter.

| | GCN | GCN+Adapter | GCN+Adapter w./ $\mathcal{L}_{task}$ and $\mathcal{L}_{task}$ | EGNN |
|---|---|---|---|---|
| Cora | 5.03 | 74.9 | 25.1 | **1.80** |
| Flickr | 37.25 | 37.6 | 29.4 | **6.34** |
| A-computers | 43.09 | 73.2 | 24.7 | **2.32** |
| A-photo | 65.08 | 85.4 | 24.8 | **2.39** |
| Coauthor-CS | 3.30 | 84.0 | 20.9 | **-0.17** |

## D.5 Ablation Study on Fine-tuning MLP

In Table 10, we present the ablation study on the effect of $\mathcal{L}_{task}$ and $\mathcal{L}_{loc}$. Specifically, we observe that after removing them, the test drawdown is significantly larger, which justify our design.

Table 10: Ablation study on the impact of MLP training procedure on editing GCN with `EGNN` in terms of test drawdown (the lower the better), here the MLP is randomly initialized.

|  | Cora | Flickr | A-computers | A-photots | Coauthor-CS |
|---|---|---|---|---|---|
| Both | **1.8** | **6.34** | **2.32** | **2.39** | -0.17 |
| Only task loss $\mathcal{L}_{task}$ | 2.34 | 8.44 | 2.88 | 7.08 | **-0.87** |
| Only locality loss $\mathcal{L}_{loc}$ | 2.57 | 10.92 | 4.63 | 4.62 | 1.01 |
| Without $\mathcal{L}_{task}$ and $\mathcal{L}_{loc}$ | 2.14 | 36.98 | 3.02 | 4.92 | 0.02 |

Here in Table 11, we also present the ablation study on the effect of initialization of MLP. Specifically, instead of random initialization, we initialize all parameters in MLP as zero. Similarly, we observe (1) after removing them, the test drawdown is significantly larger, which justify our design. (2) after fine-tuning the MLP with both loss, the zero-initialized MLP performs significantly worse than randomly initialized MLP in terms of test drawdown.

Table 11: Ablation study on the impact of MLP training procedure on editing GCN with `EGNN` in terms of test drawdown (the lower the better), here the MLP is zero-initialized.

|  | Cora | Flickr | A-computers | A-photots | Coauthor-CS |
|---|---|---|---|---|---|
| Both | **3.70** | **9.07** | **2.37** | **3.98** | 1.45 |
| Only task loss $\mathcal{L}_{task}$ | 4.98 | 11.3 | 3.19 | 4.71 | 1.22 |
| Only locality loss $\mathcal{L}_{loc}$ | 6.20 | 9.41 | 4.63 | 4.62 | 1.01 |
| Without $\mathcal{L}_{task}$ and $\mathcal{L}_{loc}$ | 6.20 | 9.79 | 4.63 | 4.62 | **1.01** |

## D.6 Why ENN performs so bad

Below we experimentally analyze why ENN performs so bad on the graph dataset. The key idea of ENN is to fine-tune the model a few steps to make it prepare for editing. Specifically, it is explicitly designed to make every sample closer to the decision boundary. In this way, the wrongly predicted samples are easier to be perturbed across the boundary. However, we found that this extra fine-tuning process significantly hurts the model performance. As shown in Table 12, we report the test accuracy for the baseline (i.e., before editing), the accuracy after fine-tuned by ENN, and the accuracy after editing. We summarize that there was a significantly accuracy drop after fine-tuned by ENN, which significantly compromises the model's performance to prepare it for editing

Table 12: The test accuracy (%) for detailed ENN performance analysis

| Model | Method | Cora | A-computers | A-photo | Coauthor-CS |
|---|---|---|---|---|---|
| GCN | Baseline | 89.4 | 87.88 | 93.77 | 94.37 |
|  | After ENN fine-tune | 32.0 | 52.97 | 9.70 | 1.92 |
|  | After Edits | 37.16 | 15.51 | 16.71 | 4.94 |
| GraphSAGE | Baseline | 86.6 | 82.83 | 94.30 | 95.17 |
|  | After ENN fine-tune | 32.00 | 7.00 | 4.60 | 13.06 |
|  | After Edits | 33.16 | 16.89 | 15.06 | 13.71 |

## D.7 Transductive Setting Evaluation

While most of our experiments are done in an inductive setting — which is specified in Section 5 — as it is often considered a harder setting than transductive, requiring the model to learn with less

information and be capable of predicting unseen nodes (Hamilton et al., 2017; Zeng et al., 2020). Our `EGNN` also work under a transductive setting, as shown in the below tables. We observe that GNNs under transductive settings also suffer catastrophic accuracy drop upon editing due to their neighbor propagation mechanism, and `EGNN` can effectively reduce it. This indicates `EGNN` is applicable and performant under the transductive setting.

Table 13: Single edit of GCN via `EGNN` under both inductive and transductive settings. We report the results in the format of Post-edit Test Accuracy (Test Drawdown).

| GCN | Cora | A-computers | A-photo | Coauthor-CS |
|---|---|---|---|---|
| GD (inductive) | 84.37 (5.03) | 44.78 (43.09) | 28.70 (65.08) | 91.07 (3.30) |
| EGNN (inductive) | **87.80 (1.80)** | **82.85 (2.32)** | **91.97 (2.39)** | **94.54 (-0.17)** |
| GD (transductive) | 86.24 (3.36) | 51.09 (36.11) | 41.49 (51.30) | 83.83 (9.13) |
| EGNN (transductive) | **87.92 (1.68)** | **85.61 (1.90)** | **90.68 (2.24)** | **93.71 (-0.73)** |

Table 14: Single edit of GraphSAGE via `EGNN` under both inductive and transductive settings. We report the results in the format of Post-edit Test Accuracy (Test Drawdown).

| **GraphSAGE** | **Cora** | **A-computers** | **A-photo** | **Coauthor-CS** |
|---|---|---|---|---|
| GD (inductive) | 82.06 (4.53) | 21.68 (61.15) | 38.98 (55.32) | 90.15 (5.01) |
| EGNN (inductive) | **85.65 (0.55)** | **84.34 (2.72)** | **92.53 (1.83)** | **95.27 (-0.01)** |
| GD (transductive) | 86.06 (3.74) | 31.15 (49.66) | 31.23 (61.17) | 88.62 (5.71) |
| EGNN (transductive) | **87.97 (1.83)** | **80.81 (2.85)** | **93.39 (0.81)** | **94.50 (-0.23)** |

## D.8 GNN Error Pattern Analysis

Here, we focus on analyzing the error pattern of 1-hop neighbors of wrongly predicted nodes (a.k.a. editing targets). We find that in most cases, although the target node is wrongly predicted, most of its close neighbors are still classified correctly, as indicated by the "bef. edit 1-hop acc" in the table below (e.g., 78.93% average 1-hop accuracy in Cora). However, if we directly edit the model to correct the prediction with vanilla gradient descent (GD), then the average accuracy of its 1-hop neighbor decreases significantly, in some cases even more than 50%.

As we analyzed before, this is mainly due to the neighbor propagation effect. In contrast, 'EGNN' has a much better 1-hop drawdown due to its propagation-free nature. Surprisingly, we found that EGNN may even greatly increase the 1-hop neighbor classification accuracy (e.g., a -7.54 drawdown on Cora). We hypothesize that this improvement occurs because the incorrectly predicted node and several of its neighbors exhibit a similar error pattern. This shared pattern can be effectively rectified through the node features alone, without the need for extensive neighborhood propagation.

Table 15: Error pattern analysis of GCN. Here "bef. edit 1-hop acc." is the average test accuracy of the 1-hop neighbors of the editing target. "GD 1-hop drawdown" and "EGNN 1-hop drawdown" are the drawdown for the 1-hop neighbors of the editing target, as influenced by the vanilla gradient descent and `EGNN` methods, respectively.

| **GCN** | **Cora** | **A-computers** | **A-photo** | **Coauthor-CS** |
|---|---|---|---|---|
| bef. edit 1-hop acc. | 78.93 | 58.10 | 83.27 | 85.58 |
| GD 1-hop drawdown | 25.37 | 33.63 | 52.02 | 33.87 |
| EGNN 1-hop drawdown | **-2.14** | **0.00** | **2.45** | **-1.90** |

## D.9 SGC and SIGN Evaluation

## E Theoretical Analysis on Why Model editing may Cry

To deeply understand why model editing may cry in GNNs, we provide a pilot theoretical analysis on one-layer GCN and one-layer MLP for binary node classification task. Specifically, we consider the model prediction be defined as $f_{\Theta}^{GCN}(\mathbf{X}) = \sigma(\tilde{\mathbf{A}}\mathbf{X}\mathbf{\Theta})$ and $f_{\Theta}^{MLP}(\mathbf{X}) = \sigma(\mathbf{X}\mathbf{\Theta})$, where $\sigma(\cdot)$

Table 16: Error pattern analysis of GraphSAGE. Here "bef. edit 1-hop acc." is the average test accuracy of the 1-hop neighbors of the editing target. "GD 1-hop drawdown" and "EGNN 1-hop drawdown" are the drawdown for the 1-hop neighbors of the editing target, as influenced by the vanilla gradient descent and EGNN methods, respectively.

| GraphSAGE | Cora | A-computers | A-photo | Coauthor-CS |
|---|---|---|---|---|
| bef. edit 1-hop acc. | 82.55 | 64.53 | 74.40 | 90.51 |
| GD 1-hop drawdown | 17.92 | 27.68 | 45.65 | 27.57 |
| EGNN 1-hop drawdown | **-7.54** | **-3.15** | **9.46** | **-2.61** |

Table 17: The results on four small scale datasets after applying one single edit for SGC and SIGN. "EGNN (raw feat.)" means the stitched MLP only take raw input features as inputs. "EGNN (prop. feat.)" means the MLP takes the propagated features as inputs where the node features are already mixed by the preprocessing step.

| | Editor | Cora | | | A-computers | | | A-photo | | | Coauthor-CS | | |
|---|---|---|---|---|---|---|---|---|---|---|---|---|---|
| | | Acc↑ | DD↓ | SR↑ | Acc↑ | DD↓ | SR↑ | Acc↑ | DD↓ | SR↑ | Acc↑ | DD↓ | SR↑ |
| SGC | GD | **83.87±4.50** | **3.73±2.50** | 1.0 | 26.17±24.29 | 60.97±26.06 | 1.0 | 24.68±15.53 | 68.53±8.44 | 1.0 | 91.75±2.47 | 3.13±1.03 | 1.0 |
| | EGNN (prop. feat) | 80.01±12.06 | 7.59±8.43 | 1.0 | 83.38±8.76 | 3.76±4.55 | 1.0 | 89.49±7.54 | 3.72±5.46 | 1.0 | 94.42±1.82 | 0.46±0.10 | 1.0 |
| | EGNN (raw feat.) | 82.12±10.36 | 5.48±11.22 | **1.0** | **84.46±9.68** | **2.68±6.48** | **1.0** | **90.87±7.28** | **2.34±3.08** | **1.0** | 94.42±2.67 | 0.46±0.16 | 1.0 |
| SIGN | GD | 82.40±3.79 | 4.80±1.52 | 1.0 | 58.33±15.05 | 27.27±12.22 | 1.0 | 66.11±16.71 | 27.53±12.00 | 1.0 | 93.98±1.26 | 1.42±0.53 | 1.0 |
| | EGNN (prop. feat) | 83.45±1.92 | 4.15±2.04 | 1.0 | 69.22±9.33 | 14.62±8.85 | 1.0 | 72.18±16.31 | 21.46±4.03 | 1.0 | 93.35±1.42 | 2.32±0.14 | 1.0 |
| | EGNN (raw feat.) | **85.36±3.23** | **2.24±0.44** | **1.0** | **83.61±1.61** | **0.22±0.13** | **1.0** | **93.08±1.22** | **0.57±0.10** | **1.0** | **95.38±0.05** | **-0.03±0.01** | 1.0 |

is sigmoid activation function, $\mathbf{X} \in \mathbb{R}^{\mathbf{n} \times \mathbf{d}}$, and $\Theta \in \mathbb{R}^{d \times 1}$. Then we have the following informal statement:

**Theorem E.1.** *For well-trained one-layer GCN $f_{\Theta_1}^{GCN}$ and one-layer MLP $f_{\Theta_2}^{MLP}$ for binary node classification task, suppose GCN has sharp locality loss landscape than MLP, model editing (parameters fine-tuning) incurs higher KL divergence locality loss for $f_{\Theta_1}^{GCN}$ than $f_{\Theta_2}^{MLP}$.*

**Remark:** Theorem E.1 represents that model editing in GNNs leads to higher prediction differences than that of MLPs. Note that such analysis is only based on one-layer model with a binary node classification task, we leave the analysis for more complicated cases (e.g., multi-layer models, and multi-class classification) for future work.

We only consider well-trained one-layer GCN and MLP for binary classification task, defined as $f_{\Theta_1}^{GCN}(\mathbf{X}) = \sigma(\tilde{\mathbf{A}}\mathbf{X}\mathbf{\Theta_1})$ and $f_{\Theta_2}^{MLP}(\mathbf{X}) = \sigma(\mathbf{X}\mathbf{\Theta_2})$, where $\sigma(\cdot)$ is sigmoid activation function, $\mathbf{X} \in \mathbb{R}^{\mathbf{n} \times \mathbf{d}}$, and $\Theta_1, \Theta_2 \in \mathbb{R}^{d \times 1}$. Define the training nodes index set as $\mathcal{V}_{train}$ and the misclassified node index as $j$, where $j \notin \mathcal{V}_{train}$. We use $\hat{y}_i$ to represent the model prediction of node $v_i$ for GCN or MLP models, and add superscript to indicate a specific model. We use cross-entropy loss for misclassified node $v_j$ in model editing and use gradient descent to update model parameters, i.e.,

$$\Theta' = \Theta - \alpha \frac{\partial \mathcal{L}_{CE}(y_j, \hat{y}_j)}{\partial \Theta}, \tag{2}$$

where $\alpha$ is step size, cross-entropy loss is $\mathcal{L}_{CE}(y_i, \hat{y}_i) = -y_i \log \hat{y}_i - (1 - y_i) \log(1 - \hat{y}_i)$. We define $\hat{y}_i'$ to represent the model prediction of node $v_i$ after model editing. We adopt the KL divergence between after and before model editing to measure the locality of the well-trained model, i.e.,

$$\mathcal{L}_{KL} = \frac{1}{|\mathcal{V}_{train}|} \sum_{i \in \mathcal{V}_{train}} \mathcal{L}_{KL}(\hat{y}_i', \hat{y}_i) = \frac{1}{|\mathcal{V}_{train}|} \sum_{i \in \mathcal{V}_{train}} \hat{y}_i' \log \frac{\hat{y}_i'}{\hat{y}_i} + (1 - \hat{y}_i') \log \frac{(1 - \hat{y}_i')}{(1 - \hat{y}_i)}, \tag{3}$$

The main goal is to compare the KL locality $\mathcal{L}_{KL}^{GCN}$ and $\mathcal{L}_{KL}^{GCN}$ for GCN and MLP model resulting from model editing with parameters update. Note that the model parameters update is relatively small, the KL locality can be effectively approximated using one-order Taylor expansion.

Note that $\mathcal{L}_{KL} = 0$ if $\Theta' = \Theta$ and model editing only leads to small model parameters perturbations, we can expand $\mathcal{L}_{KL}$ as follows:

$$\mathcal{L}_{KL} = \frac{1}{|\mathcal{V}_{train}|} \sum_{i \in \mathcal{V}_{train}} \frac{\partial \mathcal{L}_{KL}(\hat{y}_i', \hat{y}_i)}{\partial \Theta'}\Big\|_{\Theta'=\Theta}(\Theta'-\Theta) + (\Theta'-\Theta)^\top \mathbf{H}\Big\|_{\Theta'=\Theta}(\Theta'-\Theta) + o(\|\Theta'-\Theta\|_F^2) \quad (4)$$

where Hessian matrix $\mathbf{H}\big\|_{\Theta'=\Theta} = \frac{\partial^2 \mathcal{L}_{KL}(\hat{y}_i', \hat{y}_i)}{\partial(\Theta')^2}\big\|_{\Theta'=\Theta'}$. We omit the term $o(\|\Theta' - \Theta\|_F)$ due small model parameter perturbations in the following analysis. Notice that the derivative of sigmoid function is $\frac{\partial \sigma(x)}{x} = \sigma(x)\big(1 - \sigma(x)\big)$, the first derivative of KL locality for the individual sample can be given as

$$\frac{\partial \mathcal{L}_{KL}(\hat{y}_i', \hat{y}_i)}{\partial \Theta'}\Big\|_{\Theta'=\Theta} = \frac{\partial \mathcal{L}_{KL}(\hat{y}_i', \hat{y}_i)}{\partial \hat{y}_i'}\Big\|_{\hat{y}_i'=\hat{y}_i} \frac{\partial \hat{y}_i'}{\partial \Theta'}\Big\|_{\Theta'=\Theta}$$

$$= \Big( \log \frac{\hat{y}_i'}{\hat{y}_i} - \log \frac{(1-\hat{y}_i')}{(1-\hat{y}_i)} \Big)\Big\|_{\hat{y}_i'=\hat{y}_i} \frac{\partial \hat{y}_i'}{\partial \Theta'}\Big\|_{\Theta'=\Theta} = \mathbf{0}. \quad (5)$$

Therefore, the main part to analyze locality loss $\mathcal{L}_{KL}$ is Hessian matrix $\mathbf{H}\big\|_{\Theta'=\Theta}$. For simplicity, we first consider MLP model, and the first derivative of KL locality for the individual sample can be given as

$$\frac{\partial \mathcal{L}_{KL}(\hat{y}_i', \hat{y}_i)}{\partial \Theta'} = \Big( \log \frac{\hat{y}_i'}{\hat{y}_i} - \log \frac{(1-\hat{y}_i')}{(1-\hat{y}_i)} \Big)\hat{y}_i'(1-\hat{y}_i')\mathbf{X}_{i,:}^\top \triangleq g(\hat{y}_i')\mathbf{X}_{i,:}^\top \quad (6)$$

It is easy to obtain that

$$\frac{\partial g(\hat{y}_i')}{\partial \hat{y}_i'}\Big\|_{\hat{y}_i'=\hat{y}_i} = (\frac{1}{\hat{y}_i'} + \frac{1}{1-\hat{y}_i'})\hat{y}_i'(1-\hat{y}_i') + \Big( \log \frac{\hat{y}_i'}{\hat{y}_i} - \log \frac{(1-\hat{y}_i')}{(1-\hat{y}_i)} \Big)(1-2\hat{y}_i')\Big\|_{\hat{y}_i'=\hat{y}_i} = 1 \quad (7)$$

Therefore, we have Hessian matrix

$$\mathbf{H}^{MLP}\big\|_{\Theta'=\Theta} = \frac{\partial^2 \mathcal{L}_{KL}^{MLP}(\hat{y}_i', \hat{y}_i)}{\partial(\Theta')^2}\big\|_{\Theta'=\Theta'} = \frac{\partial g(\hat{y}_i')}{\partial \hat{y}_i'}\hat{y}_i'(1-\hat{y}_i')\mathbf{X}_{i,:}^\top\mathbf{X}_{i,:}$$

$$= \hat{y}_i'(1-\hat{y}_i')\mathbf{X}_{i,:}^\top\mathbf{X}_{i,:} \quad (8)$$

The locality of the well-trained MLP model for individual node $v_i$ is approximately given by

$$\mathcal{L}_{KL}(\hat{y}_i', \hat{y}_i) = (\Theta'-\Theta)^\top \mathbf{H}\big\|_{\Theta'=\Theta}(\Theta'-\Theta) = \hat{y}_i'(1-\hat{y}_i')\|\mathbf{X}_{i,:}(\Theta'-\Theta)\|^2. \quad (9)$$

Note that cross-entropy loss for misclassified node $v_j$ is adopted in model editing and model parameters update via gradient descent, ground-truth $y_j$ is given by $y_j = -u(\hat{y}_j - 0.5)$, where $u(\cdot)$ is a step function, and $\frac{\partial \mathcal{L}_{CE}(y_j, \hat{y}_j)}{\partial \hat{y}_j} = -\frac{y_j}{\hat{y}_j} + \frac{1-y_j}{1-\hat{y}_j} = \frac{u(\hat{y}_j-0.5)}{\min\{\hat{y}_j, 1-\hat{y}_j\}}$, the model editing gradient is given by

$$\frac{\partial \mathcal{L}_{CE}(y_j, \hat{y}_j)}{\partial \Theta} = \frac{u(\hat{y}_j-0.5)}{\min\{\hat{y}_j, 1-\hat{y}_j\}}\hat{y}_j(1-\hat{y}_j)\mathbf{X}_{j,:}^\top$$

$$= u(\hat{y}_j-0.5)\max\{\hat{y}_j, 1-\hat{y}_j\}\mathbf{X}_{j,:}^\top \quad (10)$$

The locality of the well-trained MLP model for individual node $v_i$ can be simplified as

$$\mathcal{L}_{KL}^{MLP}(\hat{y}_i', \hat{y}_i) = \hat{y}_i'(1-\hat{y}_i')\max\{\hat{y}_j, 1-\hat{y}_j\}\langle \mathbf{X}_{i,:}, \mathbf{X}_{j,:}\rangle \quad (11)$$

The average locality of the well-trained MLP model for training nodes is

$$\mathcal{L}_{KL}^{MLP} = \frac{1}{|\mathcal{V}_{train}|} \sum_{i \in \mathcal{V}_{train}} \hat{y}_i'(1-\hat{y}_i')\max\{\hat{y}_j, 1-\hat{y}_j\}\langle \mathbf{X}_{i,:}, \mathbf{X}_{j,:}\rangle \quad (12)$$

As for GCN model, the only difference from MLP is node feature aggregation. The average locality of the well-trained GCN model for training nodes can be obtained by replacing $\mathbf{X}$ with $\tilde{\mathbf{A}}\mathbf{X}$, i.e.,

$$\mathcal{L}_{KL}^{GCN} = \frac{1}{|\mathcal{V}_{train}|} \sum_{i \in \mathcal{V}_{train}} \hat{y}_i'(1-\hat{y}_i')\max\{\hat{y}_j, 1-\hat{y}_j\}\langle [\tilde{\mathbf{A}}\mathbf{X}]_{i,:}, [\tilde{\mathbf{A}}\mathbf{X}]_{j,:}\rangle \quad (13)$$

On the other hand, neighborhood aggregation leads node features more similar. According to (Oono & Suzuki, 2020, Proposition 1), suppose graph data has $M$ connected components and $\lambda_1 \leq \cdots \leq \lambda_n$ is the eigenvalue of $\tilde{\mathbf{A}}$ sorted in ascending order, then we have $-1 < \lambda_1, \lambda_{n-M} < 1$, and $\lambda_{n-M+1} = \cdots = \lambda_n = 1$. We mainly focus on the largest less-than-one eigenvalue defined as $\lambda \triangleq \max\limits_{k=1,\cdots,n-M} |\lambda_k| < 1$. Additionally, define subspace $\mathcal{M} \subseteq \mathbb{R}^{n \times d}$ be the linear subspace where all row vectors are equivalent, the over-smoothing issue can be measured using the distance between node feature matrix $\mathbf{X}$ and subspace $\mathcal{M}$ by $d_{\mathcal{M}}(\mathbf{X}) \triangleq \inf\{\|\mathbf{X} - \mathbf{Y}\|_F | \mathbf{Y} \in \mathcal{M}\}$. Based on (Oono & Suzuki, 2020, Theorem 2) and $\lambda < 1$, we have

$$d_{\mathcal{M}}(\tilde{\mathbf{A}}\mathbf{X}) \leq \lambda d_{\mathcal{M}}(\mathbf{X}) < d_{\mathcal{M}}(\mathbf{X}), \tag{14}$$

Note that if the raw vector of $\mathbf{Y}$ is the average row vector of $\mathbf{X}$, the distance between node feature matrix $\mathbf{X}$ and subspace $\mathcal{M}$ is given by

$$d_{\mathcal{M}}(\mathbf{X}) = \sum_{i=1}^{n} \|\mathbf{X}_{i,:} - \frac{1}{n}\sum_{i=1}^{n}\mathbf{X}_{i,:}\|_F = \sum_{i=1}^{n} \|\frac{1}{n}(\mathbf{X}_{i,:} - \sum_{k \neq i}\mathbf{X}_{k,:})\|_F$$

$$= \frac{1}{n^2}\left(\sum_{i=1}^{n} \|\mathbf{X}_{i,:}\|_F - \sum_{i \neq j}\langle\mathbf{X}_{i,:}, \mathbf{X}_{j,:}^{\top}\rangle\right), \tag{15}$$

Note that the adjacency matrix is normalized and the scale of the node features matrix is the same, i.e., $\|\mathbf{X}\|_F \approx \|\tilde{\mathbf{A}}\mathbf{X}\|_F$. Therefore, the distance between node feature matrix $\mathbf{X}$ and subspace $\mathcal{M}$ is inversely related to node feature inner products. Based on Eqs (12), (13), and (14), we have $\mathcal{L}_{KL}^{MLP} < \mathcal{L}_{KL}^{GCN}$, i.e., editable training in one-layer GCN leads to higher prediction differences than that of one-layer MLP.

