# OpenReview forum: "Editable Graph Neural Network for Node Classifications"
_ICLR.cc/2024/Conference — Submitted to ICLR 2024_

### Official Review · Reviewer_7JjA · 2023-10-31

**Soundness:** 2 fair
**Presentation:** 3 good
**Contribution:** 1 poor
**Rating:** 3
**Confidence:** 4

**Summary:**

This paper tackles the graph model editing problem by inserting an MLP in GNN architecture and freezing the GNN model during the model editing phase to alleviate the propagation of misinformation by the message-passing mechanism.

**Strengths:**

1. The problem setting is interesting.
2. The motivation example in Section 3.1 is intuitive.

**Weaknesses:**

1. My major concern regarding this paper is the performance of the proposed method. Though Tables 2, 3, and 4 show great improvement compared with GD and ENN, the actual improvement is still limited. Based on the experimental results of GCN and GraphSAGE without any graph editing on these datasets (e.g., the performance of GCN reported in [SUGRL](https://ojs.aaai.org/index.php/AAAI/article/view/20748) Tables 1 and 2), GCN achieves 91.6\% accuracy on A-Photo dataset and 84.5\% accuracy on A-Computers dataset. However, in the proposed method, EGNN achieves 91.97\% accuracy on the A-Photo dataset and 82.85\% accuracy on the A-Computers dataset. Comparing these two results, if editing the graph model could not improve the performance, then why do we need it? Thus, the authors should also report the performance of GCN and GraphSAGE without any editing in the experiment. It's important to demonstrate that graph editing can indeed improve the performance of GNN. If EGNN is worse than the vanilla GCN and GraphSAGE, then the proposed method is meaningless.

2. The novelty of this paper is somehow limited. The proposed method simply incorporates the MLP into the GNN architecture to address the issue of message-passing during the graph editing.

3. In the related work, the period is missing for the sentence "This assumption might not hold well for graph data, given the fundamental node interactions that occur during neighborhood propagation".

4. The evaluation regarding the metric DrawDown is questionable. See question 2.

**Questions:**

1. In the experiment, the authors evaluate EGNN under the inductive setting.  As many types of GNNs benefit from leveraging the feature and the graph topology of both labeled and unlabeled nodes, can EGNN work in the transductive setting?

2. In Table 2, the authors report the DrawDown of EGNN, which is the **mean absolute difference** between test accuracy before and after performing an edit. Then, how can it be a negative value (e.g., -0.17 for EGNN with GCN and -0.01 for EGNN with GraphSAGE on the Coauthor-CS dataset)?

3. In Algorithm 1 for the EGNN edit procedure, should the condition of the while loop ($\hat{y}\neq y_v$) be $\hat{y}\neq y_e$? Correct me if I misunderstand it.

4. Figure 2 suggests that the more editing it conducts, the worse performance it achieves shown in the last three subfigures, e.g., Figures GCN (Reddit), GraphSAGE (ogb-arxiv), and GraphSAGE (Reddit). When the number of sequential edits increases from 10 to 40 or 20 to 40, we can observe the DrawDown consistently increases. In this case, why do we need to edit the model anymore?

---

> ### Author Response · Authors · 2023-11-19
> **Thanks & Initial Response to Reviewer 7JjA (1/3)**
>
> We appreciate the time and effort the reviewer has dedicated to reviewing our paper, as well as finding our problem setting to be interesting and well-motivited. In this response, we aim to address the concerns and questions you have raised.
>
>
> ### **[W1 - What is the point of doing model editing if an edited model's overall acc is similar to an unedited graph? — Because the edited model is guaranteed to be correct on the edited case, which can be really important.]**
>
> For a short answer, this is because while it is true that some edited and unedited models might have similar overall accuracy readings, **the edited model is guaranteed to produce the correct output on the edited case. Where this previously wrongly predicted, now edited-to-be-correct case can be more important than some other wrong predictions.** Model editing serves to efficiently alter the model's behavior without having to retrain the model.
>
> For a long and more informative answer, we provide the below justification for the model editing task with a rich set of examples and analogies. Given the reviewer is skeptical about the editing task, we encourage you to pay this a quick scan — THANKS!
>
> ---
>
> Multiple reviewers challenge the validity of the model editing task. We believe this is mainly because we are the first to study this problem under a GNN context; where most GNN scholars might find its foreign. The motivation for editing is mainly two-fold: (1) **to address the reality that two wrong outputs may have drastically different impact/consequences**; (2) high-profile failure cases, **often manifest in a streaming manner after the initial model development** (e.g., training), but during the actual user-facing deployment. Model editing serves as a way to **timely and efficiently deliver a guaranteed patch** for those post-hoc discovered high-profile errors.
>
> ---
>
> To elaborate (1), we note that metrics like overall accuracy is, of course, meaningful, but in the real world, **not all wrong predictions are made equal. As some mistakes will inherently be more damaging than others**. Under a CV or NLP context, this can be **the difference between misclassifying a car brand and misclassifying a street-crossing child in front of a self-driving car.** For an LLM-powered chatbot, this can be the difference between citing an author of a paper wrongly and giving criminal advice to its clients.
>
> While graph data is often less intuitive than text and images, graph learning has undoubtedly been applied in many high-stake scenarios. Imagine if a graph learning algorithm is constantly giving out toxic products as antidotes in a drug synthesis scenario (vs producing a harmless placebo in a different color than expected); or recommending the profile of the abuser to the victim under a social network context (vs recommending a stranger as my highschool classmate); or, as **implied by the real-world FBI Internet Crime Report 2020**, failing to prevent an elderly citizen from losing her life saving due to internet scams [1] (vs failing to prevent me getting scammed $5 on eBay for buying a counterfeit IKEA shark.)
>
> **In the eye of the test accuracy, they are equal. But in the real world, the resulting consequences are drastically different** — and this observation motivates the model editing problem: how can one correct some high-stakes mistakes while keeping the edited model usable in all general cases (without expensive operations like retraining everything)?
>
> ---
> To elaborate on (2), model maintenance is a critical part of model lifecycle, where failure cases may appear in a streaming manner after the initial development and during the deployment (a good example is how OpenAI patches all those GPT jailbreaks). A successful model edit may serve as a guaranteed way to patch those high-profile undesired outputs without destroying the general capability of the model. Yet, it can be done in an efficient (vs retraining) and timely manner — e.g., **`EGNN` can conclude a GraphSAGE edit within 400ms ([Table 4](https://openreview.net/pdf?id=Ti0kjqFx7D#page=9)), making it the best candidate for a "hot-fix" type of job.**
>
> ---
> Finally, we would also like to note that
>
> ### **the validity of the model editing problem is well recognized in multiple domains, such as CV [2] and NLP [3, 4, 5, 6], with a [proven publishing record](https://github.com/zjunlp/KnowledgeEditingPapers).**
>
> In fact, many work even advanced from improving the numerical editing performance to investigating why some editing-specific phenomenon exists with new series of tools [7, 8]; where the graph community is arguably late with our work being the first to study this problem. **We are absolutely confident that model editing is a problem that deserves studying under a GNN context by the graph learning community.**
>
> With the added real-world example above and in [Section 1](https://openreview.net/pdf?id=Ti0kjqFx7D), we hope our reviewers may appreciate its hardly refutable importance, too.

---

> ### Author Response · Authors · 2023-11-19
> **Initial Response to Reviewer 7JjA (2/3)**
>
> ### **[W1 - The authors should also report the performance of GCN and GraphSAGE without any editing — That would just be *Post-edit Acc + Test Drawdown*, which we have already reported.]**
>
> Test drawdown is defined as decreased accuracy before and after editing (thus, smaller the better). So, **the accuracy of any model without editing applied would just be the *post-edit acc + test drawdown***, where we have already reported both metrics, and our reporting of *test drawdown* should facilitate a direct acc gap comparison.
>
>
> ### **[W2 - Technical novelty of the proposed method is limited: Even if this is true, as the first to study model editing under GNN, our main novelty contribution lies in clearly identifying, locating, and improving upon this new but important problem.]**
>
>
> We believe the novelty of a paper can be roughly viewed from two aspects: ***empirical novelty*** — as if the paper unveils properties and behavior that are unknown to the general audience (e.g., lottery ticket hypothesis [9] and emergent abilities [10]); and ***technical novelty*** — as if the propose solution designs anything new to achieve its goal (e.g., RWKV [11] and LoRA [12]).
>
> **We argue that our work provides massive contributions on the front of empirical novelty.** As before us, no one has studied the GNN model editing problem — which is a refutably important task justified in our [response to [W1]](https://openreview.net/forum?id=Ti0kjqFx7D&noteId=8v5p2LDgfk) above — let alone provided any mitigation.
>
> Our work is the ***first*** to introduce this task under a GNN context, ***first*** to reveal the fact that GNNs are naturally not robust to editing, ***first*** to locate the root of this phenomenon to be the node aggregation mechanism, and ***first*** to custom design a simple but effect approach to massively improve the editing performance (with **up to 50%** less test drawdown achieved).
>
> We acknowledge that, based on the observations by [Reviewer `7JjA`](https://openreview.net/forum?id=Ti0kjqFx7D&noteId=JBodkdw7v0), our parallel MLP design may appear straightforward and not particularly novel. However, **the way we land on this solution**, the underlying analysis, as well as how we design the *MLP training procedure* concerning $L_\text{loc}$ and $L_\text{task}$ **are non-trivial to figure out.** We kindly direct the reviewer's attention to  [Appendix D.4](https://openreview.net/pdf?id=Ti0kjqFx7D#page=16), [D.5](https://openreview.net/pdf?id=Ti0kjqFx7D#page=17), where we demonstrate many designs that resemble ours (e.g., adaptor-tuning) that end up being undesirable in a model editing context, and [Appendix E](https://openreview.net/pdf?id=Ti0kjqFx7D#page=20) for the theoretical justification. **We respectfully argue that principally developing a simple but effective solution should not discount our work's contribution.**
>
> ---
>
>
>
> [1] According to [FBI Internet Crime Report 2020]( https://www.ic3.gov/Media/PDF/AnnualReport/2020_IC3Report.pdf), around 66% of the tech support fraud victims are over 60 years; yet, they are bearing at least 84% of the total losses (>$116 million). This suggests senior citizens are more likely to experience a severe financial setback due to being the victim of the said crime, making them a prioritized focus for a proper fraud protection system. This real-world example perfectly illustrates the fact that while predicting two different potential fraud victims is considered equal under some metrics valuing *overall performance*, the difference in real-life impact can be drastic.
>
> [2] Sintsin & Plokhotnyuk & Pyrkin et al., Editable Neural Networks. ICLR 2022
> [3] De Cao et al., Editing Factual Knowledge in Language Models. EMNLP 2021.
> [4] Mitchell et al., Memory-Based Model Editing at Scale. ICML 2022
> [5]] Mitchell et al., Fast Model Editing at Scale. ICLR 2022
> [6] Zhong & Wu et al., MQUAKE: Assessing Knowledge Editing in Language Models via Multi-Hop Questions. EMNLP 2023
> [7] Meng & Bau et al., Locating and Editing Factual Associations in GPT. NeurIPS 2022
> [8] Hase et al., Does Localization Inform Editing? Surprising Differences in Causality-Based Localization vs. Knowledge Editing in Language Models. NeurIPS 2023
> [9] Frankle & Carbin, The Lottery Ticket Hypothesis: Finding Sparse, Trainable Neural Networks, ICLR 2019
> [10] Wei et al., Emergent Abilities of Large Language Models. TMLR 2022
> [11] Peng & Alcaide & Anthony et al., RWKV: Reinventing RNNs for the Transformer Era. arXiv 2023
> [12] Hu & Shen et al., LoRA: Low-Rank Adaptation of Large Language Models. arXiv 2021

---

> ### Author Response · Authors · 2023-11-19
> **Initial Response to Reviewer 7JjA (3/3)**
>
> ### **[W3 & W4/Q2 & Q3 - Typos — Thank you for the meticulous read. Those are typos, and we have now fixed them!]**
>
> The reviewer is correct that the *"mean absolute difference"* description of test drawdown is wrong; the *"absolute"* part should be removed. A negative test drawdown means the overall accuracy is improved after editing.
>
> On the note of [Algorithm 1](https://openreview.net/pdf?id=Ti0kjqFx7D#page=5), you are again correct that the RHS of the inequality should be $y_e$. We also indeed miss a period "." in the [sentence](https://openreview.net/pdf?id=Ti0kjqFx7D#page=6) you mentioned. We have fixed all typos, and we really appreciate the reviewer for doing such a close read.
>
>
> ### **[Q1 - Can EGNN work in the transductive setting? Yes! Here are some supportive results]**
>
> Because the procedure of `EGNN` does not alter the architecture of the GNN part, but only stitching a parallel MLP alongside it. As long as the model is able to make classification predictions on a certain node(s), `EGNN` can then perform model edits on such certain node(s).
>
> Our initial submission has, in fact, already included some transductive experiments. e.g., SGC in [Table 3](https://openreview.net/pdf?id=Ti0kjqFx7D#page=7). Here, we provide the following results under a transductive learning framework.
>
> > Single edit of GCN and GraphSAGE via `EGNN` under both inductive and transductive settings. We report the results in the format of ***Post-edit Test Acc. (Test Drawdown)***.
>
> |    GCN  Acc(Drawdown)            | Cora             | A-computers       | A-photo          | Coauthor-CS       |
> |---------------------|------------------|-------------------|------------------|-------------------|
> | GD (inductive)      | 84.37 (5.03)     | 44.78 (43.09)     | 28.70 (65.08)    | 91.07 (3.30)      |
> | `EGNN` (inductive)    | **87.80 (1.80)**     | **82.85 (2.32)**      | **91.97 (2.39)**     | **94.54 (-0.17)**     |
> | GD (transductive)   | 86.24 (3.36)     | 51.09 (36.11)     | 41.49 (51.30)    | 83.83 (9.13)      |
> | `EGNN` (transductive) | **87.92 (1.68)**     | **85.61 (1.90)**     | **90.68 (2.24)**     | **93.71 (-0.73)**     |
>
> |   GraphSAGE Acc(Drawdown)         | Cora             | A-computers       | A-photo          | Coauthor-CS       |
> |----------------------|------------------|-------------------|------------------|-------------------|
> | GD (inductive)       | 82.06 (4.53)     | 21.68 (61.15)     | 38.98 (55.32)    | 90.15 (5.01)      |
> | `EGNN` (inductive)     | **85.65 (0.55)**     | **84.34 (2.72)**      | **92.53 (1.83)**     | **95.27 (-0.01)**     |
> | GD (transductive)     |   86.06 (3.74)         | 31.15 (49.66)     | 31.23 (61.17)    | 88.62 (5.71)      |
> | `EGNN` (transductive)   | **87.97 (1.83)**| **80.81 (2.85)**      | **93.39 (0.81)**     | **94.50 (-0.23)**     |
>
>
> We observe that GNNs under transductive settings also suffer catastrophic accuracy drop upon editing due to their neighbor propagation mechanism, and `EGNN` can effectively reduce it. This indicates **`EGNN` is applicable and performant under the transductive setting.**
>
>
>
>
> ### **[Q4 - Why do we need to conduct model editing when the more editing it conducts, the worse performance it achieves? — We don't, those results are just there to show the limit of our method.]**
>
> We don't quite grasp the essence of the question asked. If the reviewer is interested in knowing ***why model editing is meaningful at all***, we believe this concern should be well-addressed with our [response to [W1]](https://openreview.net/forum?id=Ti0kjqFx7D&noteId=8v5p2LDgfk) above.
>
> Suppose the reviewer is interested in knowing ***why the test drawdown is increasing with more edited instances.*** In that case, this is **because the task becomes harder** when we need to ensure more and more edits are successful while retraining the overall performance of the model (as compared to just one of a few edits). To the best of our knowledge, this "performance decrease with more edits" phenomenon is commonly observed under the model editing realm (e.g., [Table 5 of [6]](https://arxiv.org/pdf/2305.14795.pdf#page=9))
>
>
> Or maybe the reviewer simply wants to know ***why we [keep on] editing the model after it already shows an unacceptable test drawdown***. Albeit some extreme cases that all those edited instances are considered absolutely necessary, **we really don't have to**; the results are just there to showcase the limit of our method, and we encourage future work with better multi-edit performance — as, after all, our goal is to introduce this task to the graph learning community, as well as sharing our related investigation findings specific to GNNs.

---

> > ### Comment · Reviewer_7JjA · 2023-11-22
> > **Response to Authors**
> >
> > Thanks for the detailed clarification. I understand graph editing could timely avoid potential loss but I am still concerned about the performance of the graph editing methods raised in W1. In W1, I mentioned that the performance of GCN and GraphSAGE without any graph editing is better than the proposed method. Although you mentioned some advantages of graph editing such as avoiding internet scams and preventing potential loss ("failing to prevent an elderly citizen from losing her life savings due to internet scams [1] (vs failing to prevent me getting scammed $5 on eBay for buying a counterfeit IKEA shark."), I can't help thinking that it may be avoided in advance by making the correct predictions in the test phase for non-graph editing methods, such as GraphSAGE as the performance of GraphSAGE is better than EGNN after graph editing.

---

> ### Author Response · Authors · 2023-11-22
> **Graph editing via EGNN is indeed not perfect, but it is a problem that is worth studying and, in many cases, the only suitable option.**
>
> We thank the reviewer for sharing insights and concerns. We think it is totally valid to be concerned about the positive test drawdown (decrease in accuracy on all test cases) shown after most GNN edit experiments. However, we argue this is not a critical (nor unique) issue for our work. We present our argument in the following two ways:
>
> ## **Editing is still a unique solution to an important problem, though with cost.**
>
> While it might be true that an unedited GraphSAGE can deliver good performance on overall metrics, **what are we going to do if this (unedited) GraphSAGE is producing a high-profile error?** The option is either sit on it and endure the potentially catastrophic losses, or address it immediately with a small cost on general performance. We believe it is rational to opt for the latter in many scenarios, where editing may come to the rescue by delivering a guaranteed patch in a timely and efficient manner.
>
> Following the intentionally absurd *"Losing life saving v.s. Losing $5 on IKEA shark"* comparison, there is no guarantee that an unedited GraphSAGE model will never wrongly predict the former case. We would even argue that there will be a lot more $5-level scams than life saving-level scams, making a slightly higher performance on overall metrics less important in a loss prevention context or anything alike.
>
> (Last, we note — though we are sure the reviewer is also well aware — that our edited notes are selected from the wrongly predicted pool of an unedited GNN. **So the reviewer's hypothesis that** *"I can't help thinking that it may be avoided in advance by making the correct predictions in the test phase for non-graph editing methods..."* **is not applicable. Because the editing target is already incorrectly classified by the unedited model at the first place.**
>
>
> ## **Research progression requires gradual improvements. Asking for a perfect performance to the first graph editing work is possibly a bit too much.**
>
> While we recognized that having a perfect editing method (0 or negative test drawdown) is for sure favorable; we emphasize that **our main novelty/contribution lies in clearly identifying and locating the graph editing problem, yet we have made massive performance improvement already** (with up to **50%** less test drawdown achieved).
>
> For context, **we would like to note that having a positive test drawdown is common among editing arts.** E.g., [6], a recent work focusing on multi-hop questions editing in NLP (also a new focus, though less establishing to NLP editing than ours' to graph editing) is seeing 21.6% and 34.2% test drawdown ([Table 3](https://arxiv.org/pdf/2305.14795.pdf#page=5)). Granted, GNN is naturally more prone to editing due to its node aggregation mechanism, **we'd respectfully argue requesting a perfect edit method as the first graph editing work is potentially too big of an ask.**
>
>
> **We argue that while having better performance is favorable, recognizing/understanding the problem is undoubtedly an essential and a more important step.** Here, we hope the reviewer can be part of building this vital recognition. Our honest opinion is that the graph community should be working on this task when there is already extensive progress on other domains [13, 14]. We hope our work can gain exposure and inspire future developments — where better test drawdown performance is surely one of the aspects — despite not being perfect performance-wise on the first try.
>
> ---
>
> [6] Zhong & Wu et al., MQUAKE: Assessing Knowledge Editing in Language Models via Multi-Hop Questions. EMNLP 2023
> [13] [KnowledgeEditingPapers | GitHub repo by zjunlp](https://github.com/zjunlp/KnowledgeEditingPapers)
> [14] [Mazzia et al.](https://arxiv.org/pdf/2310.19704.pdf), A Survey on Knowledge Editing of Neural Networks, arXiv 2023

---

### Official Review · Reviewer_fgMm · 2023-10-31

**Soundness:** 2 fair
**Presentation:** 3 good
**Contribution:** 3 good
**Rating:** 5
**Confidence:** 4

**Summary:**

This paper explorea a and important problem, i.g., GNNs model editing for node classification. This paper empirically shows that the vanilla model editing method may not perform well due to node aggregation. Furthermore, this paper proposes EGNN to correct misclassified samples while preserving other intact nodes, via stitching a trainable MLP. In this way, the power of GNNs for prediction and the editing-friendly MLP can be integrated together in EGNN.

**Strengths:**

1. The authors can leverage the GNNs’ structure learning meanwhile avoiding the spreading edition errors to guarantee the overall node classification task.

2. The experimental results validate the solution which could address all the erroneous samples.

3.  Via freezing GNNs’ part, EGNN is scalable to address misclassified nodes in the million-size graphs.

**Weaknesses:**

1. The motivation is not clear. Since the authors trained the model on a subgraph containing only the training node, how node aggregation in GNNs will spread the editing effect throughout the whole graph?

2. The experiment setting is not clear. The authors trained the model on a subgraph containing only the training node, then what graph is used for editing?

**Questions:**

See weakness.

---

> ### Author Response · Authors · 2023-11-19
> **Thanks & Initial Response to Reviewer fgMm (1/1)**
>
> We thank the reviewer for recognizing the performance and scalability of our proposed method. Here, we address your raised concerns and questions.
>
>
> ### **[W1 & W2 - If a model is trained on a subgraph containing only training nodes, what graph is used for editing? And how is the editing effect spread out the whole graph? — Just like most other inductive graph leaning arts, a test node is unseen during training, then the whole graph is used for testing.]**
>
> Both of the reviewer's questions are about the difference between ***transductive* and *inductive learning* of GNNs**. Although both frameworks are considered common knowledge to the graph learning community, we still provide a quick walkthrough of both frameworks to ensure an aligned understanding, at the risk of being redundant; then, we will talk about how model editing can integrate into both frameworks.
>
> Under a **transductive** learning setting, we feed the whole graph as training input but with the test nodes' labels masked. In this case, the **test nodes' structure is already exposed** to the model during the training phase, so we can just make predictions based on the learned node embeddings.
>
> Under an **inductive** learning setting, we follow classic work like GraphSAGE [1] to split the whole graph into **multiple disjoint subgraphs**. We denote a portion of them to be train subgraphs, with the rest being test subgraphs. In this case, the model only trains on the train subgraphs and learns a set of aggregator functions. Then, during test time, the whole graph (seen train subgraphs and unseen test subgraphs) is fed as input, where test node embeddings are learned via the aggregation functions to then make predictions.
>
> ---
>
> So, to answer the reviewer's question directly: **the whole graph is used for testing, where certain unseen test nodes within this whole graph are edited.** Without `EGNN`, should we want to edit a wrongly predicted test node to be correct, **a backpropagation accounting this prediction change will be applied on the GNN, thus spreading the editing effect to the whole graph.**
>
> ---
> While most of our experiments are done in an inductive setting — which is specified in [Section 5.1](https://openreview.net/pdf?id=Ti0kjqFx7D#page=6) — as it is often considered a harder setting than transductive, requiring the model to learn with less information and be capable of predicting unseen nodes [1, 2, 3]; **Our proposed `EGNN` can also work under a transductive setting**. We provide the following additional results:
>
>
> > Single edit of GCN and GraphSAGE via `EGNN` under both inductive and transductive settings. We report the results in the format of ***Post-edit Test Acc. (Test Drawdown)***.
>
> |    GCN  Acc(Drawdown)            | Cora             | A-computers       | A-photo          | Coauthor-CS       |
> |---------------------|------------------|-------------------|------------------|-------------------|
> | GD (inductive)      | 84.37 (5.03)     | 44.78 (43.09)     | 28.70 (65.08)    | 91.07 (3.30)      |
> | `EGNN` (inductive)    | **87.80 (1.80)**     | **82.85 (2.32)**      | **91.97 (2.39)**     | **94.54 (-0.17)**     |
> | GD (transductive)   | 86.24 (3.36)     | 51.09 (36.11)     | 41.49 (51.30)    | 83.83 (9.13)      |
> | `EGNN` (transductive) | **87.92 (1.68)**     | **85.61 (1.90)**     | **90.68 (2.24)**     | **93.71 (-0.73)**     |
>
> |   GraphSAGE Acc(Drawdown)         | Cora             | A-computers       | A-photo          | Coauthor-CS       |
> |----------------------|------------------|-------------------|------------------|-------------------|
> | GD (inductive)       | 82.06 (4.53)     | 21.68 (61.15)     | 38.98 (55.32)    | 90.15 (5.01)      |
> | `EGNN` (inductive)     | **85.65 (0.55)**     | **84.34 (2.72)**      | **92.53 (1.83)**     | **95.27 (-0.01)**     |
> | GD (transductive)     |   86.06 (3.74)         | 31.15 (49.66)     | 31.23 (61.17)    | 88.62 (5.71)      |
> | `EGNN` (transductive)   | **87.97 (1.83)**| **80.81 (2.85)**      | **93.39 (0.81)**     | **94.50 (-0.23)**     |
>
>
> We observe that GNNs under transductive settings also suffer catastrophic accuracy drop upon editing due to their neighbor propagation mechanism, and `EGNN` can effectively reduce it. This indicates `EGNN` is applicable and performant under the transductive setting.
>
>
>
> ---
> [1] Hamilton and Ying et al., Inductive Representation Learning on Large Graphs. NeurIPS 2017
> [2] Zeng & Zhou et al., GraphSAINT: Graph Sampling Based Inductive Learning Method. ICLR 2020
> [3] Chiang et al., Cluster-GCN: An Efficient Algorithm for Training Deep and Large Graph Convolutional Networks. KDD 2019

---

### Official Review · Reviewer_5Af8 · 2023-11-04

**Soundness:** 1 poor
**Presentation:** 3 good
**Contribution:** 2 fair
**Rating:** 5
**Confidence:** 4

**Summary:**

The paper presents a work on tackling the conflict between message aggregation in GNNs and model editing by stitching a MLP as auxiliary; It also provides sufficient experiment results to support the effectiveness of their method.

**Strengths:**

Strength:

•	The paper claims to be the first work studying model editing on GNNs.

•	They choose to utilize loss landscape to demonstrate the reason of accuracy drops after editing, providing a visualization of the potential rationale; a theoretical analysis in appendix also offers more solid explanation.

•	Seemingly good results on benchmarks.

**Weaknesses:**

Weakness:

* Adding an additional MLP to compensate the original model is an interesting idea. However I have big concerns about the soundness of Algorithm 1. For example, if we require all parameters of the MLP to be zero, it is almost an optimum to make L_loc and L_task lowest, especially if the pretrained GNN model already has an MLP component (e.g. using skip connections of jumping knowledge).

* The motivation of editing GNN models is not clear enough. The motivation starts from mitigating errors of GNNs in real world application; however, it does not present an example of such error or mention why the cost of wrong decision is high.

* Lack of analysis of why there exists errors in GNNs’ classification; are GNNs only making wrong prediction with certain type of inputs?

* For section 5.3, it really confuses me how the described experiments help to answer R2: it corrects the flipping nodes and then test whether model makes right prediction on same class? If I get the point correctly, the flipped nodes only take 10% of the training node, so 90% of the time model should be able to learn the correct classification, then why it  can indicate the generalization ability of EGNN?

**Questions:**

Please refer to weakness.

---

> ### Author Response · Authors · 2023-11-19
> **Thanks & Initial Response to Reviewer 5Af8 (1/3)**
>
> We thank the reviewer for recognizing the novelty of our work, the performance of our proposed method, and considering our multi-channel investigation/explanation to be solid. Here, we address your raised concerns and questions.
>
>
> ### **[W1 - Why not just make all MLP weights zero for minimal $L_\text{loc}$ and $L_\text{task}$? Because w/o such losses and random initialization, an edit may result in a massive text drawdown on non-edited instances.]**
>
> Thank you for asking this great question. We agree with the reviewer, as it is true that the $L_\text{loc}$ and $L_\text{task}$ are almost minimized with a zero-initialized MLP, albeit some trivial cases. However, if we skip this *MLP training procedure* with the two losses (as defined in [Algorithm 1](https://openreview.net/pdf?id=Ti0kjqFx7D#page=5)) and go straight to the *EGNN edit procedure* with a zero-initialized MLP — where the GNN is frozen and only the MLP weights are updatable to learn the edited instances — the **zero-initialized MLP will likely overfit to this edited instance and, therefore, causing a significant performance decrease on all test cases** (a.k.a. large test drawdown).
>
> Below, we present the ablation study on the effect of initialization of MLP. Instead of random initialization, we initialize all parameters in MLP as zero. The row of *"Without task and locality loss"* is the exact setup the reviewer described.
>
> > *Test Drawdown* of zero-initialized MLP + GCN (lower is better).
>
> |                   | Cora    | Flickr | A-computers | A-photots | Coauthor-CS |
> |-------------------|---------|--------|-------------|-----------|-------------|
> | Both              | **3.70**| **9.07**| **2.37**    | **3.98**  | 1.45        |
> | Only task loss  | 4.98 | 11.3 | 3.19 | 4.71 | 1.22 |
> | Only locality loss  | 6.20 | 9.41 | 4.63 | 4.62 | 1.01 |
> | Without task and locality loss | 6.20 | 9.79 | 4.63 | 4.62 | **1.01** |
>
> After removing task and locality loss, the test drawdown is larger, which justify our design.
>
> **We also conducted ablation studies by comparing randomly initialized to zero-initialized MLP.** We kindly direct the reviewer's attention to [Table 11](https://openreview.net/pdf?id=Ti0kjqFx7D#page=17). We observe that after fine-tuning the MLP with both loss, **the zero-initialized MLP performs significantly worse than randomly initialized MLP in terms of test drawdown.**
>
> This result supports the validity of our $L_\text{loc}$ and $L_\text{task}$ designs, as training the MLP a few iteration pre-editing will likely make it more aligned with the trained GNN, thus reduce the "catastrophic altering" when performing the editing procedure later.
>
> ### **[W4 - Why is editing a GNN trained on 10% poisoned data a proper experiment setting for RQ2 (generalizability of `EGNN`)? Because even the model is learning on 90% of the correct class data, it has undesired pre-editing class accuracy, which can be largely fixed by doing just one single-instance edit.]**
>
> Our *RQ2* is proposed as *"Can the edits generalize to correct the model prediction on other similar inputs?"* To answer the question, we need a trained GNN that is **making multiple wrong classification predictions on similar inputs** — where a GNN model trained with 10% of one of its group/class's data's labels being noisy/poisoned provides us exactly that, as such model would perform badly on this particular class. We are interested in finding if we can correct multiple wrong predictions by using `EGNN` to edit *only one single* wrongly predicted instance.
>
>
> While it is true that *"90% of the time model should be able to learn the correct classification"* as the reviewer understood, **a GNN trained on this slightly noisy-labeled data is going to have an undesired performance on the poisoned class.** Yet, with one single `EGNN` edit, we can improve the accuracy of this class without hurting the all-class accuracy much. The results in [Figure 3](https://openreview.net/pdf?id=Ti0kjqFx7D#page=7) evidenced this: it can be observed that there is a significant improvement in *Subgroup Acc* before and after a single edit. Yet, this mass correction happens without hurting much on the (all-class) *Overall Accuracy*.

---

> ### Author Response · Authors · 2023-11-19
> **Initial Response to Reviewer 5Af8 (2/3)**
>
> ### **[W2 - The motivation for editing GNN models is not clear enough. A real-world example is needed — Sure! Model editing is a well-recognized task, here we give some real life example with a senior citizen losing her life savings.]**
>
> Multiple reviewers ([R`5Af8`](https://openreview.net/forum?id=Ti0kjqFx7D&noteId=oTJO3Y4Dhc) and [R`7JjA`](https://openreview.net/forum?id=Ti0kjqFx7D&noteId=JBodkdw7v0)) challenge the validity of the model editing task. We believe this is mainly because we are the first to study the model editing problem under a GNN context; thus, most GNN scholars might find the motivation for editing foreign. We would like to borrow this chance to elaborate a bit here.
>
> The motivation for editing is mainly two-fold: (1) **to address the reality that two wrong outputs may have drastically different impact/consequences**; (2) high-profile failure cases, **often manifest in a streaming manner after the initial model development** (e.g., training), but during the actual user-facing deployment. Model editing serves as a way to **timely and efficiently deliver a guaranteed patch** for those post-hoc discovered high-profile errors.
>
> ---
>
> To elaborate (1), we note that metrics like accuracy across all test cases are, of course, meaningful, but in the real world, **not all wrong predictions are made equal. As some mistakes will inherently be more damaging than others**. Under a CV or NLP context, this can be **the difference between misclassifying a car brand and misclassifying a street-crossing child in front of a self-driving car.** For an LLM-powered chatbot, this can be the difference between citing an author of a paper wrongly and giving criminal advice to its clients.
>
> While graph data is often less intuitive than text and images, graph learning has undoubtedly been applied in many high-stake scenarios. Imagine if a graph learning algorithm is constantly giving out toxic products as antidotes in a drug synthesis scenario (vs producing a harmless placebo in a different color than expected); or recommending the profile of the abuser to the victim under a social network context (vs recommending a stranger as my highschool classmate); or, as **implied by the real-world FBI Internet Crime Report 2020**, failing to prevent an elderly citizen from losing her life saving due to internet scams [1] (vs failing to prevent me getting scammed $5 on eBay for buying a counterfeit IKEA shark.)
>
> **In the eye of the test accuracy, they are equal. But in the real world, the resulting consequences are drastically different** — and this observation motivates the model editing problem: how can one correct some high-stakes mistakes while keeping the edited model usable in all general cases (without expensive operations like retraining everything)?
>
> ---
> To elaborate on (2), model maintenance is a critical part of model lifecycle, where failure cases may appear in a streaming manner after the initial development and during the deployment (a good example is how OpenAI patches all those GPT jailbreaks). A successful model edit may serve as a guaranteed way to patch those high-profile undesired outputs without destroying the general capability of the model. Yet, it can be done in an efficient (vs retraining) and timely manner — e.g., **`EGNN` can conclude a GraphSAGE edit within 400ms ([Table 4](https://openreview.net/pdf?id=Ti0kjqFx7D#page=9)), making it the best candidate for a "hot-fix" type of job.**
>
> ---
> Finally, we would also like to note that
>
> ### **the validity of the model editing problem is well recognized in multiple domains, such as CV [2] and NLP [3, 4, 5, 6], with a [proven publishing record](https://github.com/zjunlp/KnowledgeEditingPapers).**
>
> In fact, many work even advanced from improving the numerical editing performance to investigating why some editing-specific phenomenon exists with new series of tools [7, 8]; where the graph community is arguably late with our work being the first to study this problem. **We are absolutely confident that model editing is a problem that deserves studying under a GNN context by the graph learning community.**
>
> With the added real-world example above and in [Section 1](https://openreview.net/pdf?id=Ti0kjqFx7D), we hope our reviewers may appreciate its hardly refutable importance, too.

---

> ### Author Response · Authors · 2023-11-19
> **Initial Response to Reviewer 5Af8 (3/3)**
>
> ### **[W3 - *Why there exist errors in GNNs' classification?* — This is possibly the ultimate question for the whole GNN field. Here, we provide an alternative attempt on error pattern analysis and emphasize that our method is motivation-agnostic.]**
>
> **This question, on its face value, is basically equivalent to asking *why GNNs' accuracy is not 100*% — which is challenging to provide a faithful yet comprehensive answer as of today (if not ever)**; and it is, in our opinion, way beyond the scope of our paper.
>
> We claim this question is out of scope because — as we believe the reviewer is also well aware —  **a GNN can make multiple wrong predictions for many different reasons.** Yet, our edited instances are selected at random (e.g., for the single edit experiments, we randomly select a wrongly predicted node to edit, and repeat this random selection a total of 50 times), so **the reasoning behind why a particular prediction is wrong does not influence our editing procedure (thus, "motivation-agnostic")**. We are not saying such information is not useful; it is just that our currently proposed method does not take advantage of it. Here, we increase the repetition count to 500, aiming to edit wrongly predicted nodes with different failure reasons behind them.
>
> > Single edit of GCN and GraphSAGE averaged over 500 independent edits reported in the format of ***Post-edit Test Acc. (Test Drawdown)***.
>
> | GCN    | Cora                | Amazon Computers    | Amazon Photo        | Coauthor CS         |
> |----------|---------------------|---------------------|---------------------|---------------------|
> | GD      | 86.24 (3.36)        | 42.43 (44.77)       | 40.01 (52.78)       | 85.54 (7.42)        |
> | `EGNN`  | **88.10 (1.50)**        | **85.07 (2.44)**        | **90.89 (2.03)**        | **93.64 (-0.66)**       |
>
> | GraphSAGE    | Cora                | Amazon Computers    | Amazon Photo        | Coauthor CS         |
> |----------|---------------------|---------------------|---------------------|---------------------|
> | GD     | 85.75 (4.05)        | 28.95 (51.86)       | 34.30 (58.10)       | 88.32 (6.01)        |
> | `EGNN` | **87.84 (1.96)**        | **78.37 (5.30)**        | **93.61 (0.59)**        | **94.46 (-0.19)**       |
>
> It can be observed that `EGNN` is still performant under this setting.
>
> ---
> Given classification errors are often model-dataset-training-instance specific, and different input can be grouped w.r.t. different criteria, providing a faithful answer to *"are GNNs only making wrong prediction with certain type of inputs?"* would require an exhaustive effort. **Here, we propose an alternative error pattern analysis as an attempt to explain why`EGNN` works, focusing on the 1-hot neighbors of wrongly predicted nodes (a.k.a. editing targets).**
>
>
> > Error pattern analysis of GCN and GraphSAGE. Here `bef. edit 1-hop acc.` is the average test accuracy of the 1-hop neighbors of the editing target. `GD 1-hop drawdown` and `EGNN 1-hop drawdown` are the drawdown for the 1-hop neighbors of the editing target, as influenced by the vanilla gradient descent and `EGNN` methods, respectively
>
> | GCN                       | Cora   | A-computers | A-photo | Coauthor-CS |
> |---------------------------|--------|-------------|---------|-------------|
> | bef. edit 1-hop acc. | 78.93  | 58.10       | 83.27   | 85.58       |
> | GD 1-hop drawdown         | 25.37  | 33.63       | 52.02   | 33.87       |
> | `EGNN` 1-hop drawdown       | **-2.14**  | **0.00**        | **2.45**    | **-1.90**       |
>
> | GraphSAGE                  | Cora   | A-computers | A-photo | Coauthor-CS |
> |----------------------------|--------|-------------|---------|-------------|
> | bef. edit 1-hop acc.  | 82.55  | 64.53       | 74.40   | 90.51       |
> | GD 1-hop drawdown          | 17.92  | 27.68       | 45.65   | 27.57       |
> | `EGNN `1-hop drawdown        | **-7.54**  | **-3.15**       | **9.46**    | **-2.61**       |
>
> We find that in most cases, although the target node is wrongly predicted, most of its close neighbors are still classified correctly, as indicated by the "bef. edit 1-hop acc" in the table below (e.g., 78.93% average 1-hop accuracy in Cora). However, if we directly edit the model to correct the prediction with vanilla gradient descent (GD), then the average accuracy of its 1-hop neighbor decreases significantly, in some cases even more than 50%.
>
> As we analyzed before, this is mainly due to the neighbor propagation effect. In contrast, `EGNN` has a much better 1-hop drawdown due to its propagation-free nature. Surprisingly, we found that EGNN may even greatly increase the 1-hop neighbor classification accuracy (e.g., a -7.54 drawdown on Cora). **We hypothesize that this improvement occurs because the incorrectly predicted node and several of its neighbors exhibit a similar error pattern.** This shared pattern can be effectively rectified through the node features alone, without the need for extensive neighborhood propagation.

---

> ### Author Response · Authors · 2023-11-19
> **Reference for the our initial response to Reviewer 5Af8**
>
> ---
>
>
> [1] According to [FBI Internet Crime Report 2020]( https://www.ic3.gov/Media/PDF/AnnualReport/2020_IC3Report.pdf), around 66% of the tech support fraud victims are over 60 years; yet, they are bearing at least 84% of the total losses (>$116 million). This suggests senior citizens are more likely to experience a severe financial setback due to being the victim of the said crime, making them a prioritized focus for a proper fraud protection system. This real-world example perfectly illustrates the fact that while predicting two different potential fraud victims is considered equal under some metrics valuing *overall performance*, the difference in real-life impact can be drastic.
>
> [2] Sintsin & Plokhotnyuk & Pyrkin et al., Editable Neural Networks. ICLR 2022
> [3] De Cao et al., Editing Factual Knowledge in Language Models. EMNLP 2021.
> [4] Mitchell et al., Memory-Based Model Editing at Scale. ICML 2022
> [5]] Mitchell et al., Fast Model Editing at Scale. ICLR 2022
> [6] Zhong & Wu et al., MQUAKE: Assessing Knowledge Editing in Language Models via Multi-Hop Questions. EMNLP 2023
> [7] Meng & Bau et al., Locating and Editing Factual Associations in GPT. NeurIPS 2022
> [8] Hase et al., Does Localization Inform Editing? Surprising Differences in Causality-Based Localization vs. Knowledge Editing in Language Models. NeurIPS 2023

---

> > ### Comment · Reviewer_5Af8 · 2023-11-23
> > **Response to authors**
> >
> > Thank you for the clarification and additional experiments. W1 was one of the major concerns of my previous review. I appreciate the effort the authors made to add those new ablation study, but it is not fully clear to me. "zero-initialized MLP will likely overfit to this edited instance and, therefore, causing a significant performance decrease on all test cases" -- that is exactly what I meant. I was not talking about the initialization, it is just weird to me if you add an MLP and make the result worse.  The "Without task and locality loss" might be similar to the setting that I mentioned, its comparison to "both" may be informative. But what is the implementation of GCN here? Does it have a skip connection module?
> >
> > As to the motivation, I am not doubting about the validity of the new graph editing task, but the clarification of the motivation in the context of graphs. The rebuttal somehow addressed it.
> > I will increase the score to 5.

---

> ### Author Response · Authors · 2023-11-23
> **Here we provide some clarifications on our end, and we hope the reviewer may clarify a bit as well.**
>
> We thank the reviewer for taking such a meticulous look at our work, and we are glad that our graph-focused motivation clarification has helped!
>
> Here, **we first clarify that our row of *"without task and locality loss"* means** we train a GNN, freeze the GNN weights, then stitch an MLP in parallel (either zero-initialized as showed in our [response to W1](https://openreview.net/forum?id=Ti0kjqFx7D&noteId=z0WMLPFZ0T), which the same as [Table 10](https://openreview.net/pdf?id=Ti0kjqFx7D#page=17); or randomly initialized as showed in [Table 11](https://openreview.net/pdf?id=Ti0kjqFx7D#page=17)), then edit the label of a (previously wrongly predicted) node, and then update the MLP weights to account the edited node. **May the reviewer please confirm if this is the setting you envisioned?**
>
> The main difference between this without two losses setup to `EGNN` is it skips the *MLP Training Procedure* defined in [Algorithm 1](https://openreview.net/pdf?id=Ti0kjqFx7D#page=5). Results in both initialization setups suggest training the MLP with both losses for a few iterations before editing is much better than without both losses / no fine-tuning. **Suppose the reviewer means to verify the validity of the two-loss design; we believe the two above-mentioned tables should have justified it.**
>
> ---
>
> Regarding the reviewer's question about our GCN's implementation, as well as the notion regarding skip connection. We clarify that **we implemented a vanilla GCN without skip connection**, with the only exception being we extend it to an inductive setting (should the reviewer be interested in the transductive evaluation, we also provide them as [Table 13 & 14](https://openreview.net/pdf?id=Ti0kjqFx7D#page=17). Given your mention of "skip connection," as well as your [initial feedback](https://openreview.net/forum?id=Ti0kjqFx7D&noteId=oTJO3Y4Dhc) mentioning "jumping knowledge":
>
> > *"especially if the pretrained GNN model already has an MLP component (e.g., using skip connections of jumping knowledge)."*
>
> **We suspect you meant JK-Nets [9]? If so, to the best of our understanding, the JK-Nets do not have an MLP module**, this is indicated in [its Figure 4](https://arxiv.org/pdf/1806.03536.pdf#page=5) and its [`forward()` function implementation](https://pytorch-geometric.readthedocs.io/en/latest/_modules/torch_geometric/nn/models/jumping_knowledge.html#JumpingKnowledge.forward) in `torch_geometric`, where only a concat, a max-polling, and an LSTM module are available for selection.
>
> Given LSTM is the only parameterized aggregator among the three, **may you be so kind to clarify if you want us to replace this LSTM module with MLP and try editing a JK-Net modified GCN (against `EGNN` modified GCN)?** Or maybe there is a specific variant of JK-Nets that includes MLP modules in a different fashion. In that case, we venture to trouble the reviewer to provide a citation to that paper for us to investigate.
>
> While we cannot provide direct comparison results pending verification of the model specifications, we can still provide some insights regarding  `EGNN` and JK-Nets: **We are fairly confident that JK-Nets (and its close variants) will not perform well on editing tasks**. Because even though JK-Nets can potentially use an LSTM or MLP as a layer aggregator to produce the final output, such an aggregator is learning upon the layer embedding of GNN, **where neighborhood aggregation has already happened.** Recall that our investigation showed that neighborhood aggregation would significantly hurt the editing performance, and we believe there will be no exception with an extra JK-Net layer aggregator. **That being said, please do let us know if you want an experimental confirmation**; we can easily swap out the LSTM module in JK-Nets with an MLP and conduct some editing experiments.
>
> ---
> (While not entirely the same, we did investigate the editing performance with GNN + some between-layer MLP adapters as [Appendix D.4](https://openreview.net/pdf?id=Ti0kjqFx7D#page=16) (following the adapter tuning paradigm in NLP [10]). `EGNN` is significantly more performant — just mentioning it here in case the reviewer is interested).
>
> Please let us know & thanks in advance.
>
> ---
>
> [9] [Xu et al.](https://arxiv.org/pdf/1806.03536.pdf), Representation Learning on Graphs with Jumping Knowledge Networks. ICML 2018
> [10] Houlsby et al., Parameter-Efficient Transfer Learning for NLP. ICML 2019

---

### Meta-Review · Area_Chair_VPUf · 2023-12-10

**Metareview:**

The paper under review introduces an Editable Graph Neural Network (EGNN) for node classification tasks. The proposed method aims to address the challenge of editing Graph Neural Networks (GNNs) without disrupting their message-passing mechanism. This is achieved by integrating a trainable Multi-Layer Perceptron (MLP) into the GNN architecture, which allows for targeted edits while preserving the overall model performance. The strengths of the paper include its novel approach to GNN model editing, the use of loss landscape visualization for theoretical support, and the presentation of promising experimental results. However, there are several concerns raised by the reviewers that need to be addressed.

Reviewer 5Af8 expressed concerns about the soundness of Algorithm 1, particularly regarding the addition of an MLP to the GNN architecture. The reviewer questioned the effectiveness of this approach, especially when the GNN model already includes MLP components. The authors' response and additional experiments did not fully alleviate these concerns. Both Reviewers 5Af8 and fgMm found the motivation behind the model editing for GNNs unclear. The authors' rebuttal provided some clarification, but it seems that a more detailed explanation of the real-world applications and the necessity of this approach would strengthen the paper. Reviewer 7JjA raised concerns about the performance of the proposed method compared to existing GNN models without editing. The reviewer suggested that the paper should include a comparison with the performance of standard GNN models like GCN and GraphSAGE without any editing to demonstrate the effectiveness of the proposed method. The authors' response did not convincingly address this issue. Reviewer 7JjA also questioned the novelty of the approach, suggesting that the integration of an MLP into the GNN architecture is somewhat straightforward. This concern about the level of innovation in the proposed method remains partially unaddressed. Several technical questions and concerns were raised, including the interpretation of certain experimental results (e.g., negative DrawDown values), and the impact of sequential edits on model performance. The authors' responses to these questions were only partially satisfactory.

In summary, while the paper tackles an interesting and relevant problem in the domain of GNNs, there are significant concerns regarding the soundness of the proposed approach, the clarity of motivation, the experimental setup, and the novelty of the contribution. The authors have made significant efforts to address these issues in their rebuttal, but further clarification and additional experiments might be necessary to fully convince the reviewers of the merits of this work. As it stands, the paper seems to be marginally below the acceptance threshold for this conference.

**Justification For Why Not Higher Score:**

In summary, while the paper tackles an interesting and relevant problem in the domain of GNNs, there are significant concerns regarding the soundness of the proposed approach, the clarity of motivation, the experimental setup, and the novelty of the contribution. The authors have made significant efforts to address these issues in their rebuttal, but further clarification and additional experiments might be necessary to fully convince the reviewers of the merits of this work. As it stands, the paper seems to be marginally below the acceptance threshold for this conference.

**Justification For Why Not Lower Score:**

N/A

---

### Decision · Program_Chairs · 2024-01-16

Reject